# Adaptive Gradient Quantization
# for Data-Parallel SGD

**Fartash Faghri**[1,2*]  **Iman Tabrizian**[1,2*]  **Ilia Markov**[3]  **Dan Alistarh**[3,4]

**Daniel M. Roy**[1,2]  **Ali Ramezani-Kebrya**[2]

[1]University of Toronto  [2]Vector Institute  [3]IST Austria  [4]NeuralMagic

faghri@cs.toronto.edu  iman.tabrizian@mail.utoronto.ca  alir@vectorinstitute.ai

## Abstract

Many communication-efficient variants of SGD use gradient quantization schemes. These schemes are often heuristic and fixed over the course of training. We empirically observe that the statistics of gradients of deep models change during the training. Motivated by this observation, we introduce two adaptive quantization schemes, ALQ and AMQ. In both schemes, processors update their compression schemes in parallel by efficiently computing sufficient statistics of a parametric distribution. We improve the validation accuracy by almost 2% on CIFAR-10 and 1% on ImageNet in challenging low-cost communication setups. Our adaptive methods are also significantly more robust to the choice of hyperparameters.

## 1 Introduction

Stochastic gradient descent (SGD) and its variants are currently the method of choice for training deep models. Yet, large datasets cannot always be trained on a single computational node due to memory and scalability limitations. Data-parallel SGD is a remarkably scalable variant, in particular on multi-GPU systems [1–10]. However, despite its many advantages, distribution introduces new challenges for optimization algorithms. In particular, data-parallel SGD has large communication cost due to the need to transmit potentially huge gradient vectors. Ideally, we want distributed optimization methods that match the performance of SGD on a single hypothetical super machine, while paying a negligible communication cost.

A common approach to reducing the communication cost in data-parallel SGD is gradient compression and quantization [4, 11–16]. In full-precision data-parallel SGD, each processor broadcasts its locally computed stochastic gradient vector at every iteration, whereas in quantized data-parallel SGD, each processor compresses its stochastic gradient before broadcasting. Current quantization

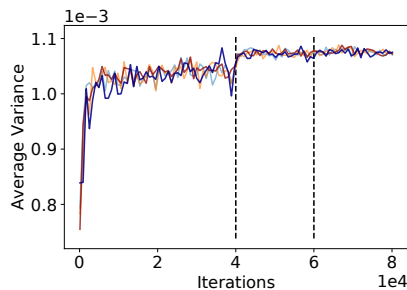

**Figure 1:** Changes in the average variance of normalized gradient coordinates in a ResNet-32 model trained on CIFAR-10. Colors distinguish different runs with different seeds. Learning rate is decayed by a factor of 10 twice at 40K and 60K iterations. The variance changes rapidly during the first epoch. The next noticeable change happens after the first learning rate drop and another one appears after the second drop.

methods are either designed heuristically or fixed prior to training. Convergence rates in a stochastic optimization problem are controlled by the trace of the gradient covariance matrix, which is referred

---

as the gradient variance in this paper [17]. As Fig. 1 shows, no fixed method can be optimal throughout the entire training because the distribution of gradients changes. A quantization method that is optimal at the first iteration will not be optimal after only a single epoch.

In this paper, we propose two adaptive methods for quantizing the gradients in data-parallel SGD. We study methods that are defined by a norm and a set of quantization levels. In Adaptive Level Quantization (ALQ), we minimize the excess variance of quantization given an estimate of the distribution of the gradients. In Adaptive Multiplier Quantization (AMQ), we minimize the same objective as ALQ by modelling quantization levels as exponentially spaced levels. AMQ solves for the optimal value of a single multiplier parametrizing the exponentially spaced levels.

## 1.1 Summary of contributions

- We propose two adaptive gradient quantization methods, ALQ and AMQ, in which processors update their compression methods in parallel.

- We establish an upper bound on the excess variance for any arbitrary sequence of quantization levels under general normalization that is tight in dimension, an upper bound on the expected number of communication bits per iteration, and strong convergence guarantees on a number of problems under standard assumptions. Our bounds hold for any adaptive method, including ALQ and AMQ.

- We improve the validation accuracy by almost $2\%$ on CIFAR-10 and $1\%$ on ImageNet in challenging low-cost communication setups. Our adaptive methods are significantly more robust to the choice of hyperparameters.[2]

## 1.2 Related work

Adaptive quantization has been used for speech communication and storage [18]. In machine learning, several biased and unbiased schemes have been proposed to compress networks and gradients. Recently, lattice-based quantization has been studied for distributed mean estimation and variance reduction [19]. In this work, we focus on unbiased and coordinate-wise schemes to compress gradients.

Alistarh et al. [20] proposed Quantized SGD (QSGD) focusing on the uniform quantization of stochastic gradients normalized to have unit Euclidean norm. Their experiments illustrate a similar quantization method, where gradients are normalized to have unit $L^\infty$ norm, achieves better performance. We refer to this method as QSGDinf or Qinf in short. Wen et al. [15] proposed TernGrad, which can be viewed as a special case of QSGDinf with three quantization levels.

Ramezani-Kebrya et al. [21] proposed nonuniform quantization levels (NUQSGD) and demonstrated superior empirical results compared to QSGDinf. Horváth et al. [22] proposed natural compression and dithering schemes, where the latter is a special case of logarithmic quantization.

There have been prior attempts at adaptive quantization methods. Zhang et al. [23] proposed ZipML, which is an optimal quantization method if all points to be quantized are known a priori. To find the optimal sequence of quantization levels, a dynamic program is solved whose computational and memory cost is quadratic in the number of points to be quantized, which in the case of gradients would correspond to their dimension. For this reason, ZipML is impractical for quantizing on the fly, and is in fact used for (offline) dataset compression. They also proposed an approximation where a subsampled set of points is used and proposed to scan the data once to find the subset. However, as we show in this paper, this one-time scan is not enough as the distribution of stochastic gradients changes during the training.

Zhang et al. [24] proposed LQ-Net, where weights and activations are quantized such that the inner products can be computed efficiently with bitwise operations. Compared to LQ-Net, our methods do not need additional memory for encoding vectors. Concurrent with our work, Fu et al. [25] proposed to quantize activations and gradients by modelling them with Weibull distributions. In comparison, our proposed methods accommodate general distributions. Further, our approach does not require any assumptions on the upper bound of the gradients.

**Input:** Local data, parameter vector (local copy) $\mathbf{w}_t$, learning rate $\alpha$, and set of update steps $\mathcal{U}$

1 **for** $t = 1$ ***to*** $T$ **do**
2     **if** $t \in \mathcal{U}$ **then**
3        **for** $i = 1$ ***to*** $M$ **do**
4           Compute sufficient statistics and update quantization levels $\boldsymbol{\ell}$;
5     **for** $i = 1$ ***to*** $M$ **do**
6        Compute $g_i(\mathbf{w}_t)$, encode $c_{i,t} \leftarrow \text{ENCODE}_{\boldsymbol{\ell}}\big(g_i(\mathbf{w}_t)\big)$, and broadcast $c_{i,t}$;
7     **for** $j = 1$ ***to*** $M$ **do**
8        Receive $c_{i,t}$ from each processor $i$ and decode $\hat{g}_i(\mathbf{w}_t) \leftarrow \text{DECODE}_{\boldsymbol{\ell}}\big(c_{i,t}\big)$;
9        Aggregate $\mathbf{w}_{t+1} \leftarrow \mathbf{P}_\Omega\big(\mathbf{w}_t - \frac{\alpha}{M} \sum_{i=1}^{M} \hat{g}_i(\mathbf{w}_t)\big)$;

**Algorithm 1:** Adaptive data-parallel SGD. Loops are executed in parallel on each machine. At certain steps, each processor computes sufficient statistics of a parametric distribution to estimate distribution of normalized coordinates.

## 2 Preliminaries: data-parallel SGD

Consider the problem of training a model parametrized by a high-dimensional vector $\mathbf{w} \in \mathbb{R}^d$. Let $\Omega \subseteq \mathbb{R}^d$ denote a closed and compact set. Our goal is to minimize $f : \Omega \to \mathbb{R}$. Assume we have access to unbiased stochastic gradients of $f$, which is $g$, such that $\mathbb{E}[g(\mathbf{w})] = \nabla f(\mathbf{w})$ for all $\mathbf{w} \in \Omega$.

The update rule for full-precision SGD is given by $\mathbf{w}_{t+1} = \mathbf{P}_\Omega\big(\mathbf{w}_t - \alpha g(\mathbf{w}_t)\big)$ where $\mathbf{w}_t$ is the current parameter vector, $\alpha$ is the learning rate, and $\mathbf{P}_\Omega$ is the Euclidean projection onto $\Omega$. We consider data-parallel SGD, which is a synchronous and distributed framework consisting of $M$ processors. Each processor receives gradients from all other processors and aggregates them. In data-parallel SGD with compression, gradients are compressed by each processor before transmission and decompressed before aggregation [20–23]. A stochastic compression method is unbiased if the vector after decompression is in expectation the same as the original vector.

## 3 Adaptive quantization

In this section, we introduce novel adaptive compression methods that adapt during the training (Algorithm 1). Let $\mathbf{v} \in \mathbb{R}^d$ be a vector we seek to quantize and $r_i = |v_i|/\|\mathbf{v}\|$ be its normalized coordinates for $i = 1, \ldots, d$.[3] Let $q_{\boldsymbol{\ell}}(r) : [0,1] \to [0,1]$ denote a random quantization function applied to the normalized coordinate $r$ using adaptable quantization levels, $\boldsymbol{\ell} = [\ell_0, \ldots, \ell_{s+1}]^\top$, where $0 = \ell_0 < \ell_1 < \cdots < \ell_s < \ell_{s+1} = 1$. For $r \in [0,1]$, let $\tau(r)$ denote the index of a level such that $\ell_{\tau(r)} \le r < \ell_{\tau(r)+1}$. Let $\rho(r) = (r - \ell_{\tau(r)})/(\ell_{\tau(r)+1} - \ell_{\tau(r)})$ be the relative distance of $r$ to level $\tau(r) + 1$. We define the random variable $h(r)$ such that $h(r) = \ell_{\tau(r)}$ with probability $1 - \rho(r)$ and $h(r) = \ell_{\tau(r)+1}$ with probability $\rho(r)$.

We define the quantization of $\mathbf{v}$ as $Q_{\boldsymbol{\ell}}(\mathbf{v}) \triangleq [q_{\boldsymbol{\ell}}(v_1), \ldots, q_{\boldsymbol{\ell}}(v_d)]^\top$ where $q_{\boldsymbol{\ell}}(v_i) = \|\mathbf{v}\| \cdot \text{sign}(v_i) \cdot h(r_i)$ and $\mathbf{h} = \{h(r_i)\}_{i=1,\ldots,d}$ are independent random variables. The encoding, $\text{ENCODE}(\mathbf{v})$, of a stochastic gradient is the combined encoding of $\|\mathbf{v}\|$ using a standard floating point encoding along with an optimal encoding of $h(r_i)$ and binary encoding of $\text{sign}(v_i)$ for each coordinate $i$. The decoding, DECODE, recovers the norm, $h(r_i)$, and the sign. Additional details of the encoding method are described in Appendix D.

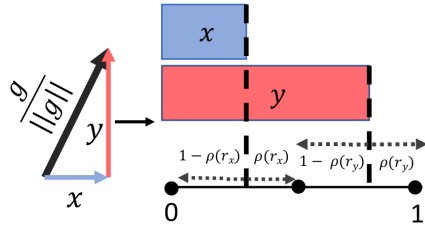

**Figure 2:** Random quantization of normalized gradient.

We define the variance of vector quantization to be the trace of the covariance matrix,

$$\mathbb{E}_{\mathbf{h}}[\|Q_{\boldsymbol{\ell}}(\mathbf{v}) - \mathbf{v}\|_2^2] = \|\mathbf{v}\|^2 \sum_{i=1}^{d} \sigma^2(r_i), \tag{1}$$

where $\sigma^2(r) = \mathbb{E}[(q_{\boldsymbol{\ell}}(r) - r)^2]$ is the variance of quantization for a single coordinate that is given by

$$\sigma^2(r) = (\ell_{\tau(r)+1} - r)(r - \ell_{\tau(r)}). \tag{2}$$

Let $\mathbf{v}$ be a random vector corresponding to a stochastic gradient and $\mathbf{h}$ capture the randomness of quantization for this random vector as defined above. We define two minimization problems, expected variance and expected normalized variance minimization:

$$\min_{\boldsymbol{\ell}\in\mathcal{L}} \mathbb{E}_{\mathbf{v},\mathbf{h}}\left[\|Q_{\boldsymbol{\ell}}(\mathbf{v})-\mathbf{v}\|_2^2\right] \quad \text{and} \quad \min_{\boldsymbol{\ell}\in\mathcal{L}} \mathbb{E}_{\mathbf{v},\mathbf{h}}\left[\|Q_{\boldsymbol{\ell}}(\mathbf{v})-\mathbf{v}\|_2^2/\|\mathbf{v}\|^2\right],$$

where $\mathcal{L} = \{\boldsymbol{\ell} : \ell_j \leq \ell_{j+1},\ \forall\, j,\ \ell_0 = 0,\ \ell_{s+1} = 1\}$ denotes the set of feasible solutions. We first focus on the problem of minimizing the expected normalized variance and then extend our methods to minimize the expected variance in Section 3.4. Let $F(r)$ denote the marginal cumulative distribution function (CDF) of a normalized coordinate $r$. Assuming normalized coordinates $r_i$ are i.i.d. given $\|\mathbf{v}\|$, the expected normalized variance minimization can be written as

$$\min_{\boldsymbol{\ell}\in\mathcal{L}} \Psi(\boldsymbol{\ell}), \text{ where } \Psi(\boldsymbol{\ell}) \triangleq \sum_{j=0}^{s}\int_{\ell_j}^{\ell_{j+1}} \sigma^2(r)\,\mathrm{d}F(r). \tag{3}$$

The following theorem suggests that solving (3) is challenging in general; however, the sub-problem of optimizing a single level given other levels can be solved efficiently in closed form. Proofs are provided in Appendix B.

**Theorem 1** (Expected normalized variance minimization). *Problem (3) is nonconvex in general. However, the optimal solution to minimize* one *level given other levels,* $\min_{\ell_i} \Psi(\boldsymbol{\ell})$, *is given by* $\ell_i^* = \beta(\ell_{i-1}, \ell_{i+1})$, *where*

$$\beta(a,c) = F^{-1}\left(F(c) - \int_a^c \frac{r-a}{c-a}\,\mathrm{d}F(r)\right). \tag{4}$$

### 3.1  ALQ: Adapting individual levels using coordinate descent

Using the single level update rule in Eq. (4) we iteratively adapt individual levels to minimize the expected normalized variance in (3). We denote quantization levels at iteration $t$ by $\boldsymbol{\ell}(t)$ starting from $t = 0$. The update rule is

$$\ell_j(t+1) = \beta(\ell_{j-1}(t), \ell_{j+1}(t)) \qquad \forall j = 1, \ldots, s. \tag{5}$$

Performing the update rule above sequentially over coordinates $j$ is a form of coordinate descent (CD) that is guaranteed to converge to a local minima. CD is particularly interesting because it does not involve any projection step to the feasible set $\mathcal{L}$. In practice, we initialize the levels with either uniform levels [20] or exponentially spaced levels proposed in [21]. We observe that starting from either initialization CD converges in small number of steps (less than 10).

### 3.2  Gradient descent

Computing $\nabla\Psi$ using Leibniz's rule [26], the gradient descent (GD) algorithm to solve (3) is based on the following update rule:

$$\ell_j(t+1) = \mathbf{P}_{\mathcal{L}}\left(\ell_j(t) - \eta(t)\frac{\partial\Psi(\boldsymbol{\ell}(t))}{\partial\ell_j}\right)$$

$$\frac{\partial\Psi(\boldsymbol{\ell}(t))}{\partial\ell_j} = \int_{\ell_{j-1}(t)}^{\ell_j(t)} (r - \ell_{j-1}(t))\,\mathrm{d}F(r) - \int_{\ell_j(t)}^{\ell_{j+1}(t)} (\ell_{j+1}(t) - r)\,\mathrm{d}F(r) \tag{6}$$

for $t = 0, 1, \ldots$ and $j = 1, \ldots, s$. Note that the projection step in Eq. (6) is itself a convex optimization problem. We propose a projection-free modification of GD update rule to systematically ensure $\boldsymbol{\ell} \in \mathcal{L}$. Let $\delta_j(t) = \min\{\ell_j(t) - \ell_{j-1}(t), \ell_{j+1}(t) - \ell_j(t)\}$ denote the minimum distance between two neighbouring levels at iteration $t$ for $j = 1, \ldots, s$. If the change in level $j$ is bounded by $\delta_j(t)/2$, it is guaranteed that $\boldsymbol{\ell} \in \mathcal{L}$. We propose to replace Eq. (6) with the following update rule:

$$\ell_j(t+1) = \ell_j(t) - \mathrm{sign}\left(\frac{\partial\Psi(\boldsymbol{\ell}(t))}{\partial\ell_j}\right)\min\left\{\eta(t)\left|\frac{\partial\Psi(\boldsymbol{\ell}(t))}{\partial\ell_j}\right|, \frac{\delta_j(t)}{2}\right\}. \tag{7}$$

### 3.3 AMQ: Exponentially spaced levels

We now focus on $\boldsymbol{\ell} = [-1, -p, \ldots, -p^s, p^s, \ldots, p, 1]^\top$, *i.e.,* exponentially spaced levels with symmetry. We can update $p$ efficiently by gradient descent using the first order derivative

$$\frac{1}{2} \frac{\mathrm{d}\Psi(p)}{\mathrm{d}p} = \int_0^{p^s} 2sp^{2s-1} \, \mathrm{d}F(r) + \sum_{j=0}^{s-1} \int_{p^{j+1}}^{p^j} \left( (jp^{j-1} + (j+1)p^j)r - (2j+1)p^{2j} \right) \mathrm{d}F(r). \quad (8)$$

### 3.4 Expected variance minimization

In this section, we consider the problem of minimizing the expected variance of quantization:

$$\min_{\boldsymbol{\ell} \in \mathcal{L}} \mathbb{E}_{\mathbf{v}, \mathbf{h}} \left[ \| Q_{\boldsymbol{\ell}}(\mathbf{v}) - \mathbf{v} \|_2^2 \right]. \quad (9)$$

To solve the expected variance minimization problem, suppose that we observe $N$ stochastic gradients $\{\mathbf{v}_1, \ldots, \mathbf{v}_N\}$. Let $F_n(r)$ and $p_n(r)$ denote the CDF and PDF of normalized coordinate conditioned on observing $\|\mathbf{v}_n\|$, respectively. By taking into account randomness in $\|\mathbf{v}\|$ and using the law of total expectation, an approximation of the expected variance in (9) is given by

$$\mathbb{E}[\| Q_s(\mathbf{v}) - \mathbf{v} \|_2^2] \approx \frac{1}{N} \sum_{n=1}^N \|\mathbf{v}_n\|^2 \sum_{j=0}^s \int_{\ell_j}^{\ell_{j+1}} \sigma^2(r) \, \mathrm{d}F_n(r). \quad (10)$$

The optimal levels to minimize Eq. (10) are a solution to the following problem:

$$\boldsymbol{\ell}^* = \arg\min_{\boldsymbol{\ell} \in \mathcal{L}} \sum_{n=1}^N \|\mathbf{v}_n\|^2 \sum_{j=0}^s \int_{\ell_j}^{\ell_{j+1}} \sigma^2(r) \, \mathrm{d}F_n(r) = \arg\min_{\boldsymbol{\ell} \in \mathcal{L}} \sum_{j=0}^s \int_{\ell_j}^{\ell_{j+1}} \sigma^2(r) \, \mathrm{d}\overline{F}(r),$$

where $\boldsymbol{\ell}^* = [\ell_1^*, \ldots, \ell_s^*]^\top$ and $\overline{F}(r) = \sum_{n=1}^N \gamma_n F_n(r)$ is the weighted sum of the conditional CDFs with $\gamma_n = \|\mathbf{v}_n\|^2 / \sum_{n=1}^N \|\mathbf{v}_n\|^2$. Note that we can accommodate both normal and truncated normal distributions by substituting associated expressions into $p_n(r)$ and $F_n(r)$. Exact update rules and analysis of computational complexity of ALQ, GD, and AMQ are discussed in Appendix C.

## 4 Theoretical guarantees

One can alternatively design quantization levels to minimize the worst-case variance. However, compared to an optimal scheme, this worst-case scheme increases the expected variance by $\Omega(d)$, which is prohibitive in deep networks. We quantify the gap in Appendix E. Proofs are in appendices.

A stochastic gradient has a *second-moment upper bound B* when $\mathbb{E}[\|g(\mathbf{w})\|_2^2] \leq B$ for all $\mathbf{w} \in \Omega$. Similarly, it has a *variance upper bound $\sigma^2$* when $\mathbb{E}[\|g(\mathbf{w}) - \nabla f(\mathbf{w})\|_2^2] \leq \sigma^2$ for all $\mathbf{w} \in \Omega$.

We consider a general adaptively quantized SGD (AQSGD) algorithm, described in Algorithm 1, where compression schemes are updated over the course of training.[4] Many convergence results in stochastic optimization rely on a variance bound. We establish such a variance bound for our adaptive methods. Further, we verify that these optimization results can be made to rely only on the average variance. In the following, we provide theoretical guarantees for AQSGD algorithm, obtain variance and code-length bounds, and convergence guarantees for convex, nonconvex, and momentum-based variants of AQSGD.

The analysis of nonadaptive methods in [20–23] can be considered as special cases of our theorems with fixed levels over the course of training. A naive adoption of available convergence guarantees results in having worst-case variance bounds over the course of training. In this paper, we show that an average variance bound can be applied on a number of problems. Under general normalization, we first obtain variance upper bound for arbitrary levels, in particular, for those obtained adaptively.

**Theorem 2** (Variance bound). *Let $\mathbf{v} \in \mathbb{R}^d$ and $q \geq 1$. The quantization of $\mathbf{v}$ under $L^q$ normalization satisfies $\mathbb{E}[Q_{\boldsymbol{\ell}}(\mathbf{v})] = \mathbf{v}$. Furthermore, we have*

$$\mathbb{E}[\| Q_{\boldsymbol{\ell}}(\mathbf{v}) - \mathbf{v} \|_2^2] \leq \epsilon_Q \|\mathbf{v}\|_2^2, \quad (11)$$

where $\epsilon_Q = \frac{(\ell_{j^*+1}/\ell_{j^*}-1)^2}{4(\ell_{j^*+1}/\ell_{j^*})} + \inf_{0<p<1} K_p \ell_1^{(2-p)} d^{\frac{2-p}{\min\{q,2\}}}$ with $j^* = \arg\max_{1\leq j \leq s} \ell_{j+1}/\ell_j$ and $K_p = \left(\frac{1}{2-p}\right)\left(\frac{1-p}{2-p}\right)^{(1-p)}$.

Theorem 2 implies that if $g(\mathbf{w})$ is a stochastic gradient with a second-moment bound $\eta$, then $Q_\ell(g(\mathbf{w}))$ is a stochastic gradient with a variance upper bound $\epsilon_Q \eta$. Note that, as long as the maximum ratio of two consecutive levels does not change, the variance upper bound decreases with the number of quantization levels. In addition, our bound matches the known $\Omega(\sqrt{d})$ lower bound in [27].

**Theorem 3** (Code-length bound). *Let $\mathbf{v} \in \mathbb{R}^d$ and $q \geq 1$. The expectation $\mathbb{E}[\|\text{ENCODE}(\mathbf{v})\|]$ of the number of communication bits needed to transmit $Q_\ell(\mathbf{v})$ under $L^q$ normalization is bounded by*

$$\mathbb{E}[\|\text{ENCODE}(\mathbf{v})\|] \leq b + n_{\ell_1,d} + d(H(L)+1) \leq b + n_{\ell_1,d} + d(\log_2(s+2)+1), \qquad (12)$$

*where $b$ is a constant, $n_{\ell_1,d} = \min\{\ell_1^{-q} + \frac{d^{1-1/q}}{\ell_1}, d\}$, $H(L)$ is the entropy of $L$ in bits, and $L$ is a random variable with the probability mass function given by*

$$\Pr(\ell_j) = \int_{\ell_{j-1}}^{\ell_j} \frac{r - \ell_{j-1}}{\ell_j - \ell_{j-1}} \, dF(r) + \int_{\ell_j}^{\ell_{j+1}} \frac{\ell_{j+1} - r}{\ell_{j+1} - \ell_j} \, dF(r)$$

*for $j = 1, \ldots, s$. In addition, we have*

$$\Pr(\ell_0 = 0) = \int_0^{\ell_1} \frac{1 - r}{\ell_1} \, dF(r) \text{ and } \Pr(\ell_{s+1} = 1) = \int_{\ell_s}^1 \frac{r - \ell_s}{1 - \ell_s} \, dF(r).$$

Theorem 3 provides a bound on the expected number of communication bits to encode the quantized stochastic gradients. As expected, the upper bound in (12) increases monotonically with $d$ and $s$.

We can combine variance and code-length upper bounds and obtain convergence guarantees for AQSGD when applied to various learning problems where we have convergence guarantees for full-precision SGD under standard assumptions.

Let $\{\boldsymbol{\ell}_1, \ldots, \boldsymbol{\ell}_K\}$ denote the set of quantization levels that AQSGD experiences on the optimization trajectory. Suppose that $\boldsymbol{\ell}_k$ is used for $T_k$ iterations with $\sum_{k=1}^K T_k = T$. For each particular $\boldsymbol{\ell}_k$, we can obtain corresponding variance bound $\epsilon_{Q,k}$ by substituting $\boldsymbol{\ell}_k$ into (11). Then the average variance upper bound is given by $\overline{\epsilon_Q} = \sum_{k=1}^K T_k \epsilon_{Q,k}/T$. For each particular $\boldsymbol{\ell}_k$, we can obtain corresponding expected code-length bound $N_{Q,k}$ by substituting random variable $L_k$ into (12). The average expected code-length bound is given by $\overline{N_Q} = \sum_{k=1}^K T_k N_{Q,k}/T$.

On convex problems, convergence guarantees can be established along the lines of [17, Theorems 6.1].

**Theorem 4** (AQSGD for nonsmooth convex optimization). *Let $f : \Omega \to \mathbb{R}$ denote a convex function and let $R^2 \triangleq \sup_{\mathbf{w} \in \Omega} \|\mathbf{w} - \mathbf{w}_0\|_2^2$. Let $\hat{B} = (1 + \overline{\epsilon_Q})B$ and $f^* = \inf_{\mathbf{w} \in \Omega} f(\mathbf{w})$. Suppose that AQSGD is executed for $T$ iterations with a learning rate $\alpha = RM/(\hat{B}\sqrt{T})$ on $M$ processors, each with access to independent stochastic gradients of $f$ with a second-moment bound $B$, such that quantization levels are updated $K$ times where $\boldsymbol{\ell}_k$ with variance bound $\epsilon_{Q,k}$ and code-length bound $N_{Q,k}$ is used for $T_k$ iterations. Then AQSGD satisfies $\mathbb{E}\left[ f\left(\frac{1}{T}\sum_{t=0}^T \mathbf{w}_t\right) \right] - f^* \leq R\hat{B}/(M\sqrt{T})$.*

*In addition, AQSGD requires at most $\overline{N_Q}$ communication bits per iteration in expectation.*

In Appendix H and Appendix I, we obtain convergence guarantees on nonconvex problems and for momentum-based variants of AQSGD under standard assumptions, respectively. Theoretical guarantees for levels with symmetry are established in Appendix J.

## 5 Experimental evaluation

In this section, we showcase the effectiveness of our adaptive quantization methods in speeding up training deep models. We compare our methods to the following baselines: single-GPU SGD (SGD), full-precision multi-GPU SGD (SuperSGD), uniform levels under $L^\infty$ normalization (QSGDinf) [20], ternary levels under $L^\infty$ normalization (TRN) [15], and exponential levels under $L^2$ normalization with exponential factor $p = 0.5$ (NUQSGD) [21, 22]. We present results for the following variations of

**Table 1:** Validation accuracy on CIFAR-10 and ImageNet using 3 bits (except for SuperSGD and TRN) with 4 GPUs.

| Quantization Method | ResNet-110 on CIFAR-10 | ResNet-32 on CIFAR-10 | ResNet-18 on ImageNet |
|---|---|---|---|
| Bucket Size | 16384 | 8192 | 8192 |
| SuperSGD | **93.86% ± 0.08** | **92.26% ± 0.04** | **68.93% ± 0.05** |
| NUQSGD [21, 22] | 84.60% ± 0.04 | 83.73% ± 0.08 | 33.36% ± 0.07 |
| QSGDinf [20] | 91.52% ± 0.07 | 89.95% ± 0.02 | 66.35% ± 0.04 |
| TRN [15] | 90.72% ± 0.06 | 89.65% ± 0.05 | 62.76% ± 0.06 |
| ALQ | **93.24% ± 0.06** | 91.30% ± 0.07 | **67.72% ± 0.07** |
| ALQ-N | **93.14% ± 0.05** | **91.96% ± 0.04** | 65.64% ± 0.07 |
| AMQ | **92.82% ± 0.04** | 91.10% ± 0.05 | 64.82% ± 0.05 |
| AMQ-N | **92.88% ± 0.02** | 91.03% ± 0.08 | 66.75% ± 0.05 |

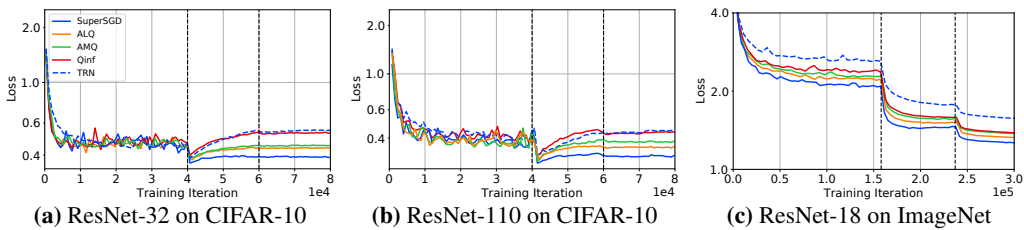

**(a)** ResNet-32 on CIFAR-10    **(b)** ResNet-110 on CIFAR-10    **(c)** ResNet-18 on ImageNet

**Figure 3: Validation loss** on CIFAR-10 and ImageNet. All methods use 3 bits except for SuperSGD and TRN. Bucket size for ResNet-110 trained on CIFAR-10 is 16384, for ResNet-32 is 8192, and for ResNet-18 on ImageNet is 8192.

our proposed methods: ALQ and AMQ (with norm adjustments in Section 3.4), and their normalized variations ALQ-N and AMQ-N (Sections 3.1 and 3.3). We present full training results on ImageNet in Appendix K along with additional experimental details.

We compare methods in terms of the number of training iterations that is independent of a particular distributed setup. In Table 1, we present results for training ResNet-32 and ResNet-110 [28] on CIFAR-10 [29], and ResNet-18 on ImageNet [30]. We simulate training with 4-GPUs on a single GPU by quantizing and dequantizing the gradient from 4 mini-batches in each training iteration. These simulations allow us to compare the performance of quantization methods to the hypothetical full-precision SuperSGD.

All quantization methods studied in this section share two hyper-parameters: the number of bits ($\log_2$ of number of quantization levels) and a bucket size. A common trick used in normalized quantization is to encode and decode a high-dimensional vector in buckets such that each coordinate is normalized by the norm of its corresponding bucket instead of the norm of the entire vector [20]. The bucket size controls the tradeoff between extra communication cost and loss of precision. With a small bucket size, there are more bucket norms to be communicated, while with a large bucket size, we lose numerical precision as a result of dividing each coordinate by a large number. In Section 5.1, we provide an empirical study of the hyperparameters.

**Matching the accuracy of SuperSGD.** Using only 3 bits (8 levels), our adaptive methods match the performance of SuperSGD on CIFAR-10 and close the gap on ImageNet (bold in Table 1). Our most flexible method, ALQ, achieves the best overall performance on ImageNet and the gap on CIFAR-10 with ALQ-N is less than 0.3%. There is at least 1.4% gap between our best performing method and previous work in training each model. To the best of our knowledge, matching the validation loss of SuperSGD has not been achieved in any previous work using only 3 bits. Fig. 3 shows the test loss and Fig. 4 shows the average gradient variance where the average is taken over gradient coordinates. Our adaptive methods successfully achieve lower variance during training.

**Comparison on the trajectory of SGD.** Fig. 5 shows the average variance on the optimization trajectory of single-GPU without quantization. This graph provides a more fair comparison of the

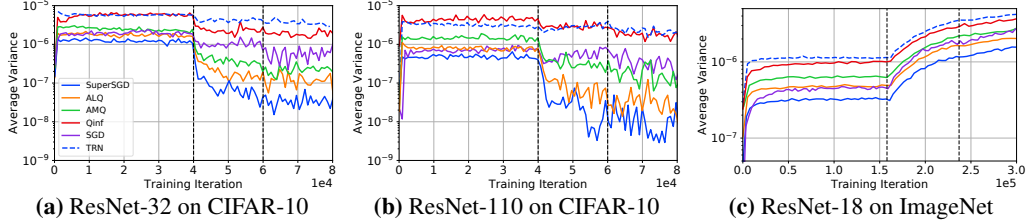

**(a)** ResNet-32 on CIFAR-10     **(b)** ResNet-110 on CIFAR-10     **(c)** ResNet-18 on ImageNet

**Figure 4: Variance** on CIFAR-10 and ImageNet. All methods use 3 bits except for SuperSGD and TRN. Bucket size for ResNet-110 trained on CIFAR-10 is 16384, for ResNet-32 is 8192, and for ResNet-18 on ImageNet is 8192.

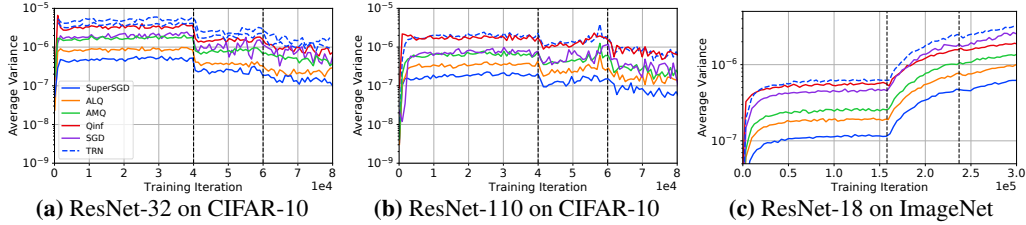

**(a)** ResNet-32 on CIFAR-10     **(b)** ResNet-110 on CIFAR-10     **(c)** ResNet-18 on ImageNet

**Figure 5: Variance (no train)** on CIFAR-10 and ImageNet. All methods use 3 bits except for SuperSGD and TRN. Bucket size for ResNet-110 trained on CIFAR-10 is 16384, for ResNet-32 is 8192, and for ResNet-18 on ImageNet is 8192.

quantization error of different methods decoupled from their impact on the optimization trajectory. ALQ effectively finds an improved set of levels that reduce the variance in quantization. ALQ matches the variance of SuperSGD on Resnet-110 (Fig. 5b). In Figs. 5b and 5c, the variance of QSGDinf is as high as TRN in the first half of training. This shows that extra levels (8 uniform levels) do not perform better unless designed carefully. As expected, the variance of SuperSGD is always smaller than the variance of SGD by a constant factor of the number of GPUs.

**Negligible computational overhead.** Our adaptive methods have similar per-step computation and communication cost compared to previous methods. On ImageNet, we save at least 60 hours from 95 hours of training and add only an additional cost of at most 10 minutes in total to adapt quantization. For bucket sizes 8192 and 16384 and 3–8 bits used in our experiments, the per-step cost relative to SuperSGD (32-bits) is 21–25% for ResNet-18 on ImageNet and 32–36% for ResNet-50. That is the same as the cost of NUQSGD and QSGDinf without additional coding or pruning with the same number of bits and bucket sizes. The cost of the additional update specific to ALQ is 0.4–0.5% of the total training time. In Appendix K.3, we provide tables with detailed timing results for varying bucket sizes and bits.

### 5.1 Hyperparameter studies

Fig. 6 shows quantization levels for each method at the end of training ResNet-32 on CIFAR-10. The quantization levels for our adaptive methods are more concentrated near zero. In Figs. 7a and 7b, we study the impact of the bucket size and number of bits on the best validation accuracy achieved by quantization methods.

**Adaptive levels are the best quantization methods across all values of bucket size and number of bits.** ALQ and ALQ-N are the best performing methods across all values of bucket size and number of bits. The good performance of ALQ-N is unexpected as it suggests quantization for vectors with different norms can be shared. In practice, ALQ-N is easier to implement and faster to update compared to ALQ. We observe a similar relation between AMQ and AMQ-N methods. Adaptive multiplier methods show inferior performance to adaptive level methods as the bucket size significantly grows (above $10^4$) or shrinks (below 100) as well as for very few bits (2). Note that there exists a known generalization gap between SGD and SuperSGD in ResNet-110 that can be closed by extensive hyperparameter tuning [31]. Our adaptive methods reduce this gap with standard hyperparameters.

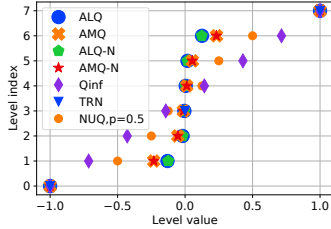

**Figure 6:** Quantization levels at the end of training ResNet-32 on CIFAR-10.

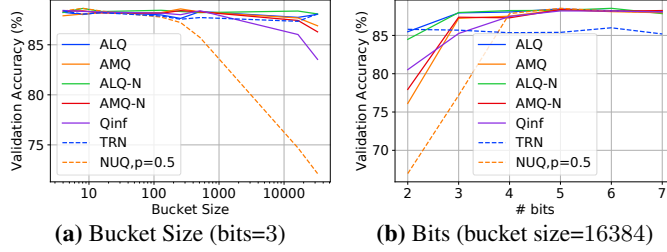

**(a)** Bucket Size (bits=3)　　　**(b)** Bits (bucket size=16384)

**Figure 7:** Effect of bucket size and number of bits on validation accuracy when training ResNet-8 on CIFAR-10

**Table 2:** Validation accuracy of ResNet32 on CIFAR-10 using 3 quantization bits (except for SuperSGD and TRN) and bucket size 16384.

| Method | 16 GPUs | 32 GPUs |
|---|---|---|
| SuperSGD | **92.17% ± 0.08** | **92.19% ± 0.04** |
| NUQSGD | 85.82% ± 0.03 | 86.36% ± 0.01 |
| QSGDinf | 89.61% ± 0.03 | 89.81% ± 0.05 |
| TRN | 88.68% ± 0.10 | 90.22% ± 0.05 |
| ALQ | **91.91% ± 0.06** | **91.89% ± 0.07** |
| ALQ-N | **92.07% ± 0.04** | **91.83% ± 0.03** |
| AMQ | 91.58% ± 0.05 | 91.38% ± 0.06 |
| AMQ-N | 91.41% ± 0.08 | 91.40% ± 0.02 |

**Bucket size significantly impacts non-adaptive methods.** For bucket size 100 and 3 bits, NUQSGD performs nearly as good as adaptive methods but quickly loses accuracy as the bucket size grows or shrinks. QSGDinf stays competitive for a wider range of bucket sizes but still loses accuracy faster than other methods. This shows the impact of bucketing as an understudied trick in evaluating quantization methods.

**Adaptive methods successfully scale to large number of GPUs.** Table 2 shows the result of training CIFAR-10 on ResNet-32 using 16 and 32 GPUs. Note that with 32 GPUs, TRN is achieving almost the accuracy of SuperSGD with only 3 quantization levels, which is expected because TRN is unbiased and the variance of aggregated gradients decreases linearly with the number of GPUs.

## 6 Conclusions

To reduce communication costs of data-parallel SGD, we introduce two adaptively quantized methods, ALQ and AMQ, to learn and adapt gradient quantization method on the fly. In addition to quantization method, in both methods, processors learn and adapt their coding methods in parallel by efficiently computing sufficient statistics of a parametric distribution. We establish tight upper bounds on the excessive variance for any arbitrary sequence of quantization levels under general normalization and on the expected number of communication bits per iteration. Under standard assumptions, we establish a number of convergence guarantees for our adaptive methods. We demonstrate the superiority of ALQ and AMQ over nonadaptive methods empirically on deep models and large datasets.

## Broader impact

This work provides additional understanding of statistical behaviour of deep machine learning models. We aim to train deep models using popular SGD algorithm as fast as possible without compromising learning outcome. As the amount of data gathered through web and a plethora of sensors deployed everywhere (e.g., IoT applications) is drastically increasing, the design of efficient machine learning algorithms that are capable of processing large-scale data in a reasonable time can improve everyone's quality of life. Our compression schemes can be used in Federated Learning settings, where a deep

model is trained on data distributed among multiple owners without exposing that data. Developing privacy-preserving learning algorithms is an integral part of responsible and ethical AI. However, the long-term impacts of our schemes may depend on how machine learning is used in society.

## Acknowledgement

The authors would like to thank Blair Bilodeau, David Fleet, Mufan Li, and Jeffrey Negrea for helpful discussions. FF was supported by OGS Scholarship. DA and IM were supported the European Research Council (ERC) under the European Union's Horizon 2020 research and innovation programme (grant agreement No 805223 ScaleML). DMR was supported by an NSERC Discovery Grant. ARK was supported by NSERC Postdoctoral Fellowship. Resources used in preparing this research were provided, in part, by the Province of Ontario, the Government of Canada through CIFAR, and companies sponsoring the Vector Institute.[5]

## Footnotes

[2]Open source code: http://github.com/tabrizian/learning-to-quantize

[3]In this section, we use $\|\cdot\|$ to denote a general $L^q$ norm with $q \ge 1$ for simplicity.

[4]Our results hold for any adaptive method, including ALQ and AMQ.

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
