[Supplementary Material · L2QNeurIPS_main_with_supp.pdf]

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

# A  CDF and its inverse

## A.1  Normal distribution

The probability density function (PDF) for $X \sim \mathcal{N}(\mu, \sigma^2)$ is defined as

$$p_{\mathcal{N}}(x) = \frac{1}{\sqrt{2\pi\sigma^2}} \exp\left(\frac{-(x-\mu)^2}{2\sigma^2}\right), \tag{13}$$

and the cumulative distribution function (CDF) defined as

$$F_{\mathcal{N}}(x) = \Phi\left(\frac{x-\mu}{\sigma}\right) = 1 - Q\left(\frac{x-\mu}{\sigma}\right) = \frac{1}{2}\left(1 + \mathrm{erf}\left(\frac{x-\mu}{\sqrt{2}\sigma}\right)\right), \tag{14}$$

where

$$\Phi(x) = \int_{-\infty}^{x} \frac{1}{\sqrt{2\pi}} \exp(-u^2/2)\,\mathrm{d}u,$$

$$Q(x) = \int_{x}^{\infty} \frac{1}{\sqrt{2\pi}} \exp(-u^2/2)\,\mathrm{d}u,$$

$$\mathrm{erf}(x) = 2\int_{0}^{x} \frac{1}{\sqrt{\pi}} \exp(-u^2)\,\mathrm{d}u.$$

The inverse of CDF for the normal distribution is given by

$$F_{\mathcal{N}}^{-1}(y) = \sigma\Phi^{-1}(y) + \mu = \sqrt{2}\sigma\,\mathrm{erf}^{-1}(2y-1) + \mu. \tag{15}$$

Various approximations of Eq. (14) and Eq. (15) are available in the literature.

## A.2  Truncated normal distribution

The probability density function (PDF) of a truncated normal distribution that lies within the interval $(a, b)$ with $-\infty < a < b < \infty$ is defined as

$$p_{\mathcal{T}}(x; a, b) = \frac{p_{\mathcal{N}}(x)}{\sigma\big(F_{\mathcal{N}}(b) - F_{\mathcal{N}}(a)\big)}, \tag{16}$$

where $p_{\mathcal{N}}(\cdot)$ is defined in Eq. (13) and the cumulative distribution function (CDF) is defined as

$$\begin{aligned} F_{\mathcal{T}}(x; a, b) &= \frac{F_{\mathcal{N}}(x) - F_{\mathcal{N}}(a)}{F_{\mathcal{N}}(b) - F_{\mathcal{N}}(a)} \\ &= \frac{\Phi((x-\mu)/\sigma) - \Phi((a-\mu)/\sigma)}{\Phi((b-\mu)/\sigma) - \Phi((a-\mu)/\sigma)}, \end{aligned} \tag{17}$$

where $F_{\mathcal{N}}(\cdot)$ and $\Phi(\cdot)$ are defined in Eq. (14). Note that the *mean* and *variance* of a random variable with truncated normal distribution are not $\mu$ and $\sigma^2$ based on our notation. The mean and variance depend on the interval $(a, b)$, which is clear in contexts that we use.

The inverse of CDF for truncated normal distribution is given by

$$F_{\mathcal{T}}^{-1}(y; a, b) = F_{\mathcal{N}}^{-1}(\overline{y}) = \sigma\Phi^{-1}(\overline{y}) + \mu, \tag{18}$$

where $\overline{y} = \big(F_{\mathcal{N}}(b) - F_{\mathcal{N}}(a)\big)y + F_{\mathcal{N}}(a)$ and $F_{\mathcal{N}}^{-1}(\cdot)$, $\Phi^{-1}(\cdot)$ are defined in Eq. (15).

# B  Expected normalized variance minimization

## B.1  Theorem 1

We prove Theorem 1 in two steps in Proposition 1 and Proposition 2.

Let $R$ denote a random variable with probability density function (PDF) $p$ and cumulative distribution function (CDF) $F$. To show that problem (3) is nonconvex, we first focus on the problem of optimizing two levels $\min_{a,b} Q(a, b)$ where

$$Q(a,b) = \int_0^a (a - r) r \, \mathrm{d}F(r) + \int_a^b (b - r)(r - a) \, \mathrm{d}F(r) + \int_b^1 (1 - r)(r - b) \, \mathrm{d}F(r) \quad (19)$$

in the range $0 \leq a \leq b \leq 1$.

**Proposition 1.** *The function $Q(a, b)$ is nonconvex in general. It becomes convex if for all $0 \leq a \leq b \leq 1$, we have*

$$b(1 - a)p(a)p(b) > (F(b) - F(a))^2. \quad (20)$$

*Proof.* Using Leibniz's rule [26], we have

$$\nabla^2 Q = \begin{bmatrix} bp(a) & F(a) - F(b) \\ F(a) - F(b) & (1 - a)p(b) \end{bmatrix}. \quad (21)$$

We can find the eigenvalues of $\nabla^2 Q$ by solving $|\nabla^2 Q - \lambda \mathbf{I}| = 0$, which leads to the following quadratic equation:

$$\lambda^2 - (bp(a) + (1 - a)p(b))\lambda + \big(b(1 - a)p(a)p(b) - (F(b) - F(a))^2\big) = 0. \quad (22)$$

We note that (20) is sufficient to guarantee $\nabla^2 Q \succeq \mathbf{0}$. $\qquad\square$

**Corollary 1.** *The sufficient condition (20) is satisfied if $R$ is uniformly distributed in the range $[0, 1]$.*

We now solve the problem of optimizing a single level, *i.e.,* $\min_b Q(b)$ where

$$Q(b) = \int_a^b (b - r)(r - a) \, \mathrm{d}F(r) + \int_b^c (c - r)(r - b) \, \mathrm{d}F(r). \quad (23)$$

**Proposition 2.** *The optimal solution to minimize $Q(b)$ is given by*[6]

$$b^* = F^{-1} \left( F(c) - \int_a^c \frac{r - a}{c - a} \, \mathrm{d}F(r) \right). \quad (24)$$

*Proof.* Using Leibniz's rule, we have

$$\frac{\mathrm{d}Q}{\mathrm{d}b} = \int_a^b (r - a) \, \mathrm{d}F(r) - \int_b^c (c - r) \, \mathrm{d}F(r),$$
$$\frac{\mathrm{d}^2 Q}{\mathrm{d}b^2} = (c - a)p(b).$$

We note that $Q$ is convex so we can find the closed-form optimal solution through satisfying the first order optimality condition. $\qquad\square$

**Corollary 2.** *In the special case with $a = 0$ and $c = 1$, the optimal solution to minimize $Q(b)$ is given by*

$$b^* = F^{-1}(1 - \mathbb{E}[R]).$$

For the special case of a truncated normal, the inner integral is evaluated as

$$\int_a^c \frac{r - a}{c - a} \, \mathrm{d}F_{\mathcal{T}}(r) = \frac{\mu - a}{c - a}(F_{\mathcal{T}}(c) - F_{\mathcal{T}}(a)) - \frac{\sigma^2}{c - a}(p_{\mathcal{T}}(c) - p_{\mathcal{T}}(a)),$$

where $F_{\mathcal{T}}$ and $p_{\mathcal{T}}$, the CDF and PDF of the truncated normal, are defined in Appendix A.

## B.2 Projected Gradient Descent

For the special case of a normal or truncated normal distribution, the gradient of the expected normalized variance used in Section 3.2 is:

$$\frac{\partial \Psi(\boldsymbol{\ell}(t))}{\partial \ell_j} = (\mu - \ell_{j-1}(t)) \left( F(\ell_j(t)) - F(\ell_{j-1}(t)) \right) + \sigma^2 \left( p(\ell_{j-1}(t)) - p(\ell_{j+1}(t)) \right)$$
$$+ (\mu - \ell_{j+1}(t)) \left( F(\ell_{j+1}(t)) - F(\ell_j(t)) \right). \tag{25}$$

## B.3 Symmetric Levels

In this section, we introduce quantization method with symmetrical levels. This is particularly useful when the estimated PDF of normalized coordinates is an even function, which is the case for normal distribution with zero mean. Let $(-\ell_{s+1}, \ldots, -\ell_1, \ell_1, \ldots, \ell_{s+1})$ denote a sequence of symmetrical quantization levels w.r.t. 0 where $0 < \ell_1 < \cdots < \ell_s < \ell_{s+1} = 1$. Let rewrite the vector of adaptable quantization levels as $\tilde{\boldsymbol{\ell}} = [\tilde{\ell}_1, \ldots, \tilde{\ell}_{2s+2}]^\top$ where $\tilde{\ell}_1 = -1 < \tilde{\ell}_2 < \cdots < \tilde{\ell}_{2s+1} < \tilde{\ell}_{2s+2} = 1$. For $\theta \in [-1, 1]$, let $\tilde{\ell}(\theta)$ and $\rho(\theta)$ satisfy $\tilde{\ell}_{\tilde{\ell}(\theta)} \le \theta \le \tilde{\ell}_{\tilde{\ell}(\theta)+1}$ and $\theta = \left(1 - \rho(\theta)\right)\tilde{\ell}_{\tilde{\ell}(\theta)} + \rho(\theta)\tilde{\ell}_{\tilde{\ell}(\theta)+1}$, respectively. Let $\mathbf{v} \in \mathbb{R}^d$ and $\theta_i = v_i/\|\mathbf{v}\|$ for $i = 1, \ldots, d$.

**Definition 1.** *The symmetrical quantization of a vector $\mathbf{v} \in \mathbb{R}^d$ is*

$$Q_{\tilde{\boldsymbol{\ell}}}(\mathbf{v}) \triangleq [q_{\tilde{\boldsymbol{\ell}}}(v_1), \ldots, q_{\tilde{\boldsymbol{\ell}}}(v_d)]^\top, \tag{26}$$

*where $q_{\tilde{\boldsymbol{\ell}}}(v_i) = \|\mathbf{v}\| \cdot h(\theta_i)$ and $h(\theta_i)$'s are independent random variables such that $h(\theta_i) = \tilde{\ell}_{\tilde{\ell}(\theta_i)}$ with probability $1 - \rho(\theta_i)$ and $h(\theta_i) = \tilde{\ell}_{\tilde{\ell}(\theta_i)+1}$ otherwise where $\rho(\theta) = (\theta - \tilde{\ell}_{\tilde{\ell}(\theta)})/(\tilde{\ell}_{\tilde{\ell}(\theta)+1} - \tilde{\ell}_{\tilde{\ell}(\theta)})$.*

Let $r_i = |\theta_i|$ for $i = 1, \ldots, d$. We have the following propositions.

**Proposition 3.** *The variance of quantization with symmetric levels is given by*

$$\mathbb{E}[\|Q_{\tilde{\boldsymbol{\ell}}}(\mathbf{v}) - \mathbf{v}\|^2] = \|\mathbf{v}\|^2 \sum_{i=1}^{d} \sigma^2(r_i), \tag{27}$$

*where*

$$\sigma^2(r_i) = \sum_{r_i \in [0, \ell_1]} (\ell_1^2 - r_i^2) + \sum_{j=1}^{s} \sum_{r_i \in [\ell_j, \ell_{j+1}]} (r_i - \ell_j)(\ell_{j+1} - r_i). \tag{28}$$

*Proof.* Note that for symmetrical levels, we have

$$\sum_{|\theta_i| \in [\ell_j, \ell_{j+1}]} (|\theta_i| - \ell_j)(\ell_{j+1} - |\theta_i|) = \sum_{\theta_i \in [-\ell_{j+1}, -\ell_j]} (\theta_i + \ell_{j+1})(-\ell_j - \theta_i)$$
$$+ \sum_{\theta_i \in [\ell_j, \ell_{j+1}]} (\theta_i - \ell_j)(\ell_{j+1} - \theta_i).$$

In addition, we have

$$\sum_{|\theta_i| \in [0, \ell_1]} (\ell_1^2 - |\theta_i|^2) = \sum_{\theta_i \in [-\ell_1, \ell_1]} (\ell_1^2 - \theta_i^2).$$

$\square$

**Proposition 4.** *If PDF of normalized gradients is an even function,i.e., $p(-\theta) = p(\theta)$ for $\theta \in [-1, 1]$, the expected normalized variance in (3) can be rewritten as*

$$\Psi(\boldsymbol{\ell}) = 2 \left( \int_0^{\ell_1} (\ell_1^2 - r^2) \, \mathrm{d}F(r) + \sum_{j=1}^{s} \int_{\ell_j}^{\ell_{j+1}} (\ell_{j+1} - r)(r - \ell_j) \, \mathrm{d}F(r) \right). \tag{29}$$

### B.3.1 GD

For the case of symmetrical levels, the gradient of the expected normalized variance is given by

$$
\begin{aligned}
\frac{1}{2}\frac{\partial \Psi(\boldsymbol{\ell}(t))}{\partial \ell_1} &= 2\ell_1(t)\big(F(\ell_1(t)) - F(0)\big) - \int_{\ell_1(t)}^{\ell_2(t)} (\ell_2(t) - r)\,\mathrm{d}F(r),\\
\frac{1}{2}\frac{\partial \Psi(\boldsymbol{\ell}(t))}{\partial \ell_j} &= \int_{\ell_{j-1}(t)}^{\ell_j(t)} (r - \ell_{j-1}(t))\,\mathrm{d}F(r) - \int_{\ell_j(t)}^{\ell_{j+1}(t)} (\ell_{j+1}(t) - r)\,\mathrm{d}F(r)
\end{aligned}
\tag{30}
$$

for $j = 2, \dots, s$.

For the special case of normal or truncated normal distribution, $\frac{1}{2}\frac{\partial \Psi(\boldsymbol{\ell}(t))}{\partial \ell_j}$ is obtained by the R.H.S. of Eq. (25) for $j = 2, \dots, s$. In addition, we have:

$$
\frac{1}{2}\frac{\partial \Psi(\boldsymbol{\ell}(t))}{\partial \ell_1} = 2\ell_1(t)\big(F(\ell_1(t)) - F(0)\big) + (\mu - \ell_2(t))\left(F(\ell_2(t)) - F(\ell_1(t))\right) - \sigma^2 p(\ell_2(t)).
$$

### B.3.2 CD

In the following lemma, we solve the problem of optimizing a single level, *i.e.*, $\min_b \tilde{Q}(b)$ where

$$
\tilde{Q}(b) = \int_0^b (b^2 - r^2)\,\mathrm{d}F(r) + \int_b^c (c - r)(r - b)\,\mathrm{d}F(r).
$$

**Proposition 5.** *The optimal solution to minimize $\tilde{Q}$ satisfies*

$$
2b^*(F(b^*) - F(0)) = \int_{b^*}^c (c - r)\,\mathrm{d}F(r),
\tag{31}
$$

*where $F$ is the CDF of the normalized coordinate.*

*Proof.* Using Leibniz's rule [26], we have

$$
\begin{aligned}
\frac{\mathrm{d}\tilde{Q}}{\mathrm{d}b} &= \int_0^b 2b\,\mathrm{d}F(r) - \int_b^c (c - r)\,\mathrm{d}F(r),\\
\frac{\mathrm{d}^2\tilde{Q}}{\mathrm{d}b^2} &= bp(b) + cp(b) + 2(F(b) - F(0)) > 0.
\end{aligned}
$$

We note that $\tilde{Q}$ is convex so we can find the unique optimal solution by satisfying the first order optimality condition. $\qquad\square$

We can solve Eq. (31) efficiently through a bisection search on $[0, \ell_2(t)]$. In particular, for the special case of normal and truncated normal, starting with a random $\boldsymbol{\ell}(0)$, the update rule at iteration $t + 1$ is the same as Eq. (5) for $j = 2, \dots, s$. For $\ell_1(t + 1)$, we solve

$$
\begin{aligned}
&(\ell_2(t) - \mu)\left(F(\ell_2(t)) - F(\ell_1(t + 1))\right) + 2\ell_1(t + 1)\left(F(0) - F(\ell_1(t + 1))\right)\\
&+ \sigma^2 \left(p(\ell_2(t)) - p(\ell_1(t + 1))\right) = 0.
\end{aligned}
$$

### B.3.3 Exponentially spaced levels

In this section, we focus on $\boldsymbol{\ell} = [-1, -p, \dots, -p^s, p^s, \dots, p, 1]^\top$, *i.e.*, exponentially spaced levels with symmetry. Following Proposition 4, the expected normalized variance is given by

$$
\Psi(p) = 2\left(\int_0^{p^s} (p^{2s} - r^2)\,\mathrm{d}F(r) + \sum_{j=0}^{s-1}\int_{p^{j+1}}^{p^j} (p^j - r)(r - p^{j+1})\,\mathrm{d}F(r)\right).
\tag{32}
$$

Using Leibniz's rule, we can compute the first order derivative:

$$\frac{1}{2}\frac{\mathrm{d}\Psi(p)}{\mathrm{d}p} = \int_0^{p^s} 2sp^{2s-1}\,\mathrm{d}F(r) + \sum_{j=0}^{s-1}\int_{p^{j+1}}^{p^j}\left((jp^{j-1}+(j+1)p^j)r - (2j+1)p^{2j}\right)\,\mathrm{d}F(r).$$

In particular, in the special case of a normal or truncated normal distribution, we have

$$\frac{1}{2}\frac{\mathrm{d}\Psi(p)}{\mathrm{d}p} = 2sp^{2s-1}\left(F(p^s)-F(0)\right) + \sigma^2\sum_{j=0}^{s-1}(jp^{j-1}+(j+1)p^j)\left(p(p^{j+1})-p(p^j)\right)$$

$$+ \sum_{j=0}^{s-1}\left(\mu(jp^{j-1}+(j+1)p^j)-(2j+1)p^{2j}\right)\left(F(p^j)-F(p^{j+1})\right).$$

We can update $p$ efficiently by a gradient descent algorithm as we have a closed-form expression to find the gradient function.

## C   Expected variance minimization in Section 3.4

In the following, we provide the update rules and the analysis of computation complexity of ALQ, GD, and AMQ.

### C.1   ALQ (CD update)

Starting with a random $\boldsymbol{\ell}(0)$, for $t = 0, 1, \ldots$ and $j = 1, \ldots, s$, we solve

$$\overline{F}(\ell_j(t+1)) = \overline{F}(\ell_{j+1}(t)) - \int_{\ell_{j-1}(t)}^{\ell_{j+1}(t)}\frac{r-\ell_{j-1}(t)}{\ell_{j+1}(t)-\ell_{j-1}(t)}\,\mathrm{d}\overline{F}(r) \tag{33}$$

by a bisection search on $[\ell_{j-1}(t), \ell_{j+1}(t)]$.

In the special case of (truncated) normal distribution, to obtain $\ell_j(t+1)$ for $j = 1, \ldots, s$, we solve

$$\sum_{n=1}^N \gamma_n\left((\ell_{j-1}(t)-\mu_n)\left(F_n(\ell_{j+1}(t))-F_n(\ell_{j-1}(t))\right)+\sigma_n^2\left(p_n(\ell_{j+1}(t))-p_n(\ell_{j-1}(t))\right)\right)$$
$$+ (\ell_{j+1}(t)-\ell_{j-1}(t))\left(\overline{F}(\ell_{j+1}(t))-\overline{F}(\ell_j(t+1))\right) = 0 \tag{34}$$

by a bisection search on $[\ell_{j-1}(t), \ell_{j+1}(t)]$.

In the special case of symmetrical levels and (truncated) normal distribution, the update rule is the same as Eq. (34) for $j = 2, \ldots, s$. For $\ell_1(t+1)$, we solve

$$\sum_{n=1}^N \gamma_n\left((\ell_2(t)-\mu_n)\left(F_n(\ell_2(t))-F_n(\ell_1(t+1))\right)+\sigma_n^2\left(p_n(\ell_2(t))-p_n(\ell_1(t+1))\right)\right)$$
$$+ 2\ell_1(t+1)\left(\overline{F}(0)-\overline{F}(\ell_1(t+1))\right) = 0 \tag{35}$$

by a bisection search on $[0, \ell_2(t)]$.

### C.2   GD update

The GD algorithm to minimize Eq. (10) is based on the following update rule by starting from a random $\boldsymbol{\ell}(0)$:

$$\ell_j(t+1) = \ell_j(t) - \mathrm{sign}(\hat{g}(t,j))\min\{\eta(t)|\hat{g}(t,j)|, \delta_j(t)/2\}$$
$$\hat{g}(t,j) = \int_{\ell_{j-1}(t)}^{\ell_j(t)}(r-\ell_{j-1}(t))\,\mathrm{d}\overline{F}(r) - \int_{\ell_j(t)}^{\ell_{j+1}(t)}(\ell_{j+1}(t)-r)\,\mathrm{d}\overline{F}(r) \tag{36}$$

for $t = 0, 1, \ldots$ and $j = 1, \ldots, s$.

In the special case of (truncated) normal distribution, we have

$$\hat{g}(t,j) = \sum_{n=1}^{N} \gamma_n \Big( (\mu_n - \ell_{j-1}(t)) \, (F_n(\ell_j(t)) - F_n(\ell_{j-1}(t)))$$

$$+ (\mu_n - \ell_{j+1}(t)) \, (F_n(\ell_{j+1}(t)) - F_n(\ell_j(t))) + \sigma_n^2 \, (p_n(\ell_{j-1}(t)) - p_n(\ell_{j+1}(t))) \Big). \tag{37}$$

### C.3 AMQ (GD update with exponentially spaced levels)

In this section, we focus on $\boldsymbol{\ell} = [-1, -p, \dots, -p^s, p^s, \dots, p, 1]^\top$, *i.e.,* exponentially spaced levels with symmetry. Following Proposition 4, the expected variance of quantization is given by

$$\tilde{V}(p) = 2 \left( \int_0^{p^s} (p^{2s} - r^2) \, \mathrm{d}\overline{F}(r) + \sum_{j=0}^{s-1} \int_{p^{j+1}}^{p^j} (p^j - r)(r - p^{j+1}) \, \mathrm{d}\overline{F}(r) \right). \tag{38}$$

In the special case of (truncated) normal distribution, we have

$$\frac{\mathrm{d}\tilde{V}(p)}{\mathrm{d}p} = +2sp^{2s-1} \left( \overline{F}(p^s) - \overline{F}(0) \right)$$

$$+ \sum_{n=1}^{N} \gamma_n \Big( \sum_{j=0}^{s-1} \left( \mu_n (jp^{j-1} + (j+1)p^j) - (2j+1)p^{2j} \right) \left( F_n(p^j) - F_n(p^{j+1}) \right)$$

$$+ \sigma_n^2 \sum_{j=0}^{s-1} (jp^{j-1} + (j+1)p^j) \left( p_n(p^{j+1}) - p_n(p^j) \right) \Big).$$

We can update $p$ efficiently by a gradient descent algorithm as we have a closed-form expression to find the gradient function.

### C.4 Computational complexity and scalability

The number of iterations for ALQ method to converge is in the order of $O(s \log(1/\epsilon))$ where $\epsilon$ is the suboptimality gap of bisection search. The number of iterations for AMQ method to achieve a local minimum with gap $\epsilon$ is $O(1/\epsilon)$. The total number of gradient computations for GD method to achieve a local minimum with gap $\epsilon$ is $O(s/\epsilon)$. Note that processors can run our methods in parallel. The time complexity of these methods is independent of the number of samples, the number of processors, and the number of parameters. The extra computational overhead is negligible compared to costs of computation of stochastic gradients and communication. Furthermore, we do not need to optimize levels at each iteration. Our experimental results suggest that it is sufficient to optimize levels at the lr_scheduler iterations.

## D  Encoding

A quantized gradient $Q_{\boldsymbol{\ell}}(\mathbf{v})$ can be uniquely determined by a tuple $(\|\mathbf{v}\|, \mathbf{s}, \mathbf{h})$ where $\|\mathbf{v}\|$ is the Euclidean norm of the gradient, $\mathbf{s} \triangleq [\mathrm{sign}(v_1), \dots, \mathrm{sign}(v_d)]^\top$ is the vector of signs of the coordinates $v_i$'s, and $\mathbf{h} \triangleq [h(r_1), \dots, h(r_d)]^\top$ are the discrete values of the normalized coordinates after quantization.

We can describe the ENCODE function (for Algorithm 1) in terms of the tuple $(\|\mathbf{v}\|, \mathbf{s}, \mathbf{h})$ and encoding/decoding scheme $\Gamma : \{\ell_0, \ell_1, \dots, \ell_{s+1}\} \to \{0,1\}^*$ and $\Gamma^{-1} : \{0,1\}^* \to \{\ell_0, \ell_1, \dots, \ell_{s+1}\}$. Any lossless prefix code can be used for encoding/decoding. In particular, we consider Huffman coding due to its efficient encoding/decoding and its optimality in terms of achieving the minimum expected code length among methods encoding symbols separately [32].

The encoding, ENCODE($\mathbf{v}$), of a stochastic gradient is as follows: We first encode the norm $\|\mathbf{v}\|$ using $b$ bits where, in practice, we use standard 32-bit floating point encoding. We then proceed in rounds, $t = 0, 1, \dots, d$. Noting that we do not need to encode the sign bit for zero entries of $\mathbf{h}$, on

round $t$, if $h(r_t) = 0$, we transmit $\Gamma(0)$. If $h(r_t) \neq 0$, we transmit $\Gamma(h_{r_t})$, transmit one bit encoding the $\text{sign}(v_t)$, and proceed to the next entry of $\mathbf{h}$.

The DECODE function (for Algorithm 1) simply reads $b$ bits to reconstruct $\|\mathbf{v}\|$. Using $\Gamma^{-1}$, it decodes the index of the first coordinate, depending on whether the decoded entry is zero or nonzero, it may read one bit indicating the sign, and then proceeds to decode the next symbol. The process proceeds in rounds, mimicking the encoding process, finishing when all coordinates have been decoded. Note that we can improve coding efficiency by encoding blocks of symbols at the cost of increasing encoding/decoding complexity. In this paper, we focus on a simple lossless prefix coding scheme that encodes symbols separately.

In order to implement an efficient lossless prefix code, we need to know the probabilities associated with our symbols to be coded, *i.e.*, $\{\ell_0, \ell_1, \ldots, \ell_{s+1}\}$. Fortunately, we can compute those probabilities using the marginal PDF of normalized coordinates and quantization levels as shown in this proposition:

**Proposition 6.** *The probability of occurrence of $\ell_j$ (weight of symbol $\ell_j$) is given by*

$$\Pr(\ell_j) = \int_{\ell_{j-1}}^{\ell_j} \frac{r - \ell_{j-1}}{\ell_j - \ell_{j-1}}\, \mathrm{d}F(r) + \int_{\ell_j}^{\ell_{j+1}} \frac{\ell_{j+1} - r}{\ell_{j+1} - \ell_j}\, \mathrm{d}F(r)$$

*for $j = 1, \ldots, s$ where $F$ is the marginal CDF of normalized coordinates. In addition, we have*

$$\Pr(\ell_0 = 0) = \int_0^{\ell_1} \frac{1 - r}{\ell_1}\, \mathrm{d}F(r) \text{ and } \Pr(\ell_{s+1} = 1) = \int_{\ell_s}^1 \frac{r - \ell_s}{1 - \ell_s}\, \mathrm{d}F(r).$$

In the special case of truncated normal distribution, we have the symbol probabilities in closed-form:

**Corollary 3.** *Suppose normalized coordinates have truncated normal distribution with PDF $p_{\mathcal{T}}$ and CDF $F_{\mathcal{T}}$ defined in Appendix A.2. The probability of occurrence of $\ell_j$ (weight of symbol $\ell_j$) is given by*

$$\Pr(\ell_j) = \frac{\sigma^2(p_{\mathcal{T}}(\ell_{j-1}) - p_{\mathcal{T}}(\ell_j)) + (\mu - \ell_{j-1})(F_{\mathcal{T}}(\ell_j) - F_{\mathcal{T}}(\ell_{j-1}))}{\ell_j - \ell_{j-1}}$$
$$+ \frac{\sigma^2(p_{\mathcal{T}}(\ell_{j+1}) - p_{\mathcal{T}}(\ell_j)) + (\ell_{j+1} - \mu)(F_{\mathcal{T}}(\ell_{j+1}) - F_{\mathcal{T}}(\ell_j))}{\ell_{j+1} - \ell_j}$$

*for $j = 1, \ldots, s$. In addition, we have*

$$\Pr(\ell_0 = 0) = \frac{\sigma^2(p_{\mathcal{T}}(\ell_1) - p_{\mathcal{T}}(0)) + (\ell_1 - \mu)(F_{\mathcal{T}}(\ell_1) - F_{\mathcal{T}}(\ell_0))}{\ell_1},$$
$$\Pr(\ell_{s+1} = 1) = \frac{\sigma^2(p_{\mathcal{T}}(\ell_s) - p_{\mathcal{T}}(1)) + (\mu - \ell_s)(F_{\mathcal{T}}(1) - F_{\mathcal{T}}(\ell_s))}{1 - \ell_s}.$$

Note that each processor can construct the Huffman tree by knowing $\boldsymbol{\ell}$ and estimating $\mu$ and $\sigma^2$. A Huffman tree of a source with $n$ symbols can be constructed in time $O(n)$ if the symbols are sorted by probability. Huffman codes are optimal in terms of expected code-length:

**Theorem 5** (Cover and Thomas 32, Theorems 5.4.1 and 5.8.1). *Let $X$ denote a random source. The expected code-length of an optimal prefix code, e.g., Huffman code to compress $X$ is bounded by $H(X) \leq \mathbb{E}[L] \leq H(X) + 1$ where $H(X)$ is the entropy of $X$ in bits.*

## E  Variance gap

**Proposition 7** (Variance gap). *For any distribution where the gap between the expected variance of a normalized coordinate under an optimal quantization to minimize (3) and the worst-case one is lower bounded by some constant, the total gap is lower bounded by $\Omega(d)$. We quantify this gap for the special case of one level with truncated normal density.*

*Proof.* Suppose we want to design a single level $b \in (0, 1)$. As shown in Corollary 2, the optimal $b$ to minimize $Q(b)$ is given by $b^* = F^{-1}(1 - \mathbb{E}[R])$. Let $R$ have truncated normal density with

parameters $\mu, \sigma^2$ in the unit interval. Plugging PDF and CDF of $R$, the optimal level to minimize (3) is given by

$$b^* = \sigma \Phi^{-1}\Big(\Delta(1-\mu) + \sigma\delta + \Phi\Big(-\frac{\mu}{\sigma}\Big)\Big) + \mu,$$

where $\Delta = \Phi((1-\mu)/\sigma) - \Phi(-\mu/\sigma)$, $\delta = \phi((1-\mu)/\sigma) - \phi(-\mu/\sigma)$, $\Phi(x) = \int_{-\infty}^{x} \exp(-u^2/2)\,\mathrm{d}u/\sqrt{2\pi}$, and $\phi(x) = \exp(-x^2/2)/\sqrt{2\pi}$.

Note that $\hat{b} = 1/2$ minimizes the worst-case variance upper bound in Eq. (1) [27]. In general, $b^* \neq \hat{b}$ depending on $\mu$ and $\sigma^2$. Without loss of generality, assume $b^* > \hat{b}$.

As show in Proposition 2, $Q(b) = \int_0^b (b-r)(r-a)\,\mathrm{d}F(r) + \int_b^1 (1-r)(r-b)\,\mathrm{d}F(r)$ is convex and

$$\frac{\mathrm{d}^2 Q}{\mathrm{d}b^2} = \frac{\phi((b-\mu)/\sigma)}{\sigma\Delta}.$$

In the interval $[\hat{b}, b^*]$, we have

$$\frac{\mathrm{d}^2 Q}{\mathrm{d}b^2} \geq \gamma \triangleq \min\Big\{\frac{\phi((b^*-\mu)/\sigma)}{\sigma\Delta}, \frac{\phi((\hat{b}-\mu)/\sigma)}{\sigma\Delta}\Big\}.$$

In this interval, $Q$ is $\gamma$-strongly convex, *i.e.,*

$$Q(\hat{b}) \geq Q(b^*) + \frac{\gamma}{2}(b^* - \hat{b})^2.$$

Hence, the gap in the expected normalized variance under $b^*$ and $\hat{b}$ is lower bounded by:

$$\frac{1}{2}d\gamma(b^* - \hat{b})^2.$$

$\square$

# F   Proof of Theorem 2 (variance bound)

Let $\boldsymbol{\ell} = [\ell_0, \ell_1, \ldots, \ell_s, \ell_{s+1}]^\top$ denote arbitrary quantization levels where $\ell_0 = 0 < \ell_1 < \cdots < \ell_{s+1} = 1$. The variance of $Q_{\boldsymbol{\ell}}(\mathbf{v})$, *i.e.,* $V_{\boldsymbol{\ell}}(\mathbf{v}) = \mathbb{E}[\|Q_{\boldsymbol{\ell}}(\mathbf{v}) - \mathbf{v}\|_2^2]$, can be expressed as

$$V_{\boldsymbol{\ell}}(\mathbf{v}) = \|\mathbf{v}\|_q^2 \Big( \sum_{r_i \in \mathcal{I}_0} (\ell_1 - r_i)r_i + \sum_{j=1}^s \sum_{r_i \in \mathcal{I}_j} (\ell_{j+1} - r_i)(r_i - \ell_j) \Big), \tag{39}$$

where $r_i = |v_i|/\|\mathbf{v}\|_q$, $\mathcal{I}_0 \triangleq [0, \ell_1]$, and $\mathcal{I}_j \triangleq [\ell_j, \ell_{j+1}]$ for $j = 1, \ldots, s$.

We can find the minimum $k_j$ that satisfies $(\ell_{j+1} - r)(r - \ell_j) \leq k_j r^2$ for $r \in \mathcal{I}_j$ and $j = 1, \ldots, s$. Expressing $r = \ell_j \theta$, we can find $k$ through solving

$$\begin{aligned} k_j &= \max_{1 \leq \theta \leq \ell_{j+1}/\ell_j} \frac{(\ell_{j+1}/\ell_j - \theta)(\theta - 1)}{\theta^2} \\ &= \frac{(\ell_{j+1}/\ell_j - 1)^2}{4(\ell_{j+1}/\ell_j)}. \end{aligned} \tag{40}$$

We note that $\ell_{j+1}/\ell_j > 1$ and $(x-1)^2/(4x)$ is monotonically increasing function of $x$ for $x > 1$.

Furthermore, note that

$$\sum_{r_i \notin \mathcal{I}_0} r_i^2 \leq \frac{\|\mathbf{v}\|_2^2}{\|\mathbf{v}\|_q^2}.$$

Substituting Eq. (40) into Eq. (39), an upper bound on $V_{\boldsymbol{\ell}}(\mathbf{v})$ is given by

$$V_{\boldsymbol{\ell}}(\mathbf{v}) \leq \|\mathbf{v}\|_q^2 \Big( \frac{(\ell_{j^*+1}/\ell_{j^*} - 1)^2}{4(\ell_{j^*+1}/\ell_{j^*})} \frac{\|\mathbf{v}\|_2^2}{\|\mathbf{v}\|_q^2} + \sum_{r_i \in \mathcal{I}_0} (\ell_1 - r_i)r_i \Big),$$

where $j^* = \arg\max_{1 \leq j \leq s} \ell_{j+1}/\ell_j$.

In our proofs, we use the following known lemma.

**Lemma 1.** *Let* $\mathbf{v} \in \mathbb{R}^d$. *Then, for all* $0 < p < q$, *we have* $\|\mathbf{v}\|_q \leq \|\mathbf{v}\|_p \leq d^{1/p-1/q}\|\mathbf{v}\|_q$.

Note that Lemma 1 holds even when $q < 1$ and $\|\cdot\|_q$ is merely a seminorm.

In the following, we derive a bound on $\sum_{r_i \in \mathcal{I}_0}(2^{-s} - r_i)r_i$, which completes the proof.

**Lemma 2.** *Let* $p \in (0,1)$ *and* $r \in \mathcal{I}_0$. *Then we have* $r(\ell_1 - r) \leq K_p \ell_1^{(2-p)}r^p$ *where*

$$K_p = \Big(\frac{1/p}{2/p - 1}\Big)\Big(\frac{1/p - 1}{2/p - 1}\Big)^{(1-p)}. \tag{41}$$

*Proof.* We can find the minimum $K_p$ through solving $K_p = \ell_1^{(-2+p)}\max_{r \in \mathcal{I}_0} r(\ell_1 - r)/r^p$. Expressing the optimization variable as $r = \ell_1 \theta^{1/p}$, $K_p$ can be obtained by solving this problem:

$$K_p = \max_{0 < \theta < 1} \theta^{1/p-1} - \theta^{2/p-1}. \tag{42}$$

We can solve (42) and obtain the optimal solution $\theta^* = \big(\frac{1/p-1}{2/p-1}\big)^p$. Substituting $\theta^*$ into (42), we obtain Eq. (41). $\square$

Let $\mathcal{S}_j$ denote the coordinates of vector $\mathbf{v}$ whose elements fall into the $(j+1)$-th bin, *i.e.,* $\mathcal{S}_j \triangleq \{i : r_i \in [l_j, l_{j+1}]\}$ for $j = 0, \ldots, s$.

Then, for any $0 < p < 1$ and $q \geq 2$, we have

$$\|\mathbf{v}\|_q^2 \sum_{r_i \in \mathcal{I}_0} r_i^p = \|\mathbf{v}\|_q^{2-p} \sum_{i \in \mathcal{S}_0} |v_i|^p$$
$$\leq \|\mathbf{v}\|_q^{2-p}\|\mathbf{v}\|_p^p$$
$$\leq \|\mathbf{v}\|_q^{2-p}\|\mathbf{v}\|_2^p d^{1-p/2}$$
$$\leq \|\mathbf{v}\|_2^2 d^{1-p/2},$$

where the third inequality holds as $\|\mathbf{v}\|_p \leq \|\mathbf{v}\|_2 d^{1/p-1/2}$ using Lemma 1 and the last inequality holds as $\|\mathbf{v}\|_q \leq \|\mathbf{v}\|_2$ for $q \geq 2$.

This gives us an upper bound on $V_{\boldsymbol{\ell}}(\mathbf{v})$:

$$V_{\boldsymbol{\ell}}(\mathbf{v}) \leq \|\mathbf{v}\|_2^2 \Big(\frac{(\ell_{j^*+1}/\ell_{j^*} - 1)^2}{4(\ell_{j^*+1}/\ell_{j^*})} + K_p \ell_1^{(2-p)} d^{1-p/2}\Big).$$

For $q \geq 1$, we have $\|\mathbf{v}\|_q^{2-p} \leq \|\mathbf{v}\|_2^{2-p} d^{\frac{2-p}{\min\{q,2\}} - \frac{2-p}{2}}$, which gives

$$V_{\boldsymbol{\ell}}(\mathbf{v}) \leq \|\mathbf{v}\|_2^2 \Big(\frac{(\ell_{j^*+1}/\ell_{j^*} - 1)^2}{4(\ell_{j^*+1}/\ell_{j^*})} + K_p \ell_1^{(2-p)} d^{\frac{2-p}{\min\{q,2\}}}\Big).$$

## G   Proof of Theorem 3 (code-length bound)

Let $|\cdot|$ denote the length of a binary string. In this section, we obtain an upper bound on $\mathbb{E}[|\mathrm{ENCODE}(\mathbf{v})|]$, *i.e.,* the expected number of communication bits per iteration. Recall from Appendix D that the quantized vectors $Q_{\boldsymbol{\ell}}(\mathbf{v})$ is uniquely determined by the tuple $(\|\mathbf{v}\|_q, \mathbf{s}, \mathbf{h})$.

We first encode the norm $\|\mathbf{v}\|_q$ using $b$ bits where, in practice, we use standard 32-bit floating point encoding.

We send one bit for each nonzero entry of $\mathbf{h}$. Let $\mathcal{S}_j \triangleq \{i : r_i \in [l_j, l_{j+1}]\}$ and $d_j \triangleq |\mathcal{S}_j|$ for $j = 0, \ldots, s$. We have an upper bound on the expected number of nonzero entries as follows:

**Lemma 3.** *Let* $\mathbf{v} \in \mathbb{R}^d$. *The expected number of nonzeros in* $Q_{\boldsymbol{\ell}}(\mathbf{v})$ *is bounded above by*

$$\mathbb{E}[\|Q_{\boldsymbol{\ell}}(\mathbf{v})\|_0] \leq \ell_1^{-q} + \frac{d^{1-1/q}}{\ell_1}.$$

*Proof.* Note that $d - d_0 \leq \ell_1^{-q}$ since

$$(d - d_0)\ell_1{}^q \leq \sum_{i \notin \mathcal{S}_0} r_i^q \leq 1. \tag{43}$$

For each $i \in \mathcal{S}_0$, $Q_{\boldsymbol{\ell}}(v_i)$ becomes zero with probability $1 - r_i/\ell_1$, which results in

$$\mathbb{E}[\|Q_{\boldsymbol{\ell}}(\mathbf{v})\|_0] \leq d - d_0 + \sum_{i \in \mathcal{S}_0} r_i/\ell_1$$

$$\leq \ell_1{}^{-q} + \frac{d^{1-1/q}}{\ell_1}, \tag{44}$$

where the last inequality holds as $\|\mathbf{v}\|_1 \leq \|\mathbf{v}\|_q d^{1-1/q}$ using Lemma 1. $\qquad \square$

For each entry of $\mathbf{h}$, we send the associated codeword. The optimal expected code-length for transmitting one random symbol is within one bit of the entropy of the source. Hence, we need to transmit upto $d(H(L) + 1)$ to transmit entries of $\mathbf{h}$ [32]. Putting everything together, we have

$$\mathbb{E}[|\text{ENCODE}(\mathbf{v})|] \leq b + n_{\ell_1, d} + d(H(L) + 1).$$

Finally, note that the entropy of a source with $n$ outcomes is bounded above by $\log_2(n)$.

# H  AQSGD for smooth nonconvex optimization

On nonconvex problems, we can establish convergence guarantees in terms of convergence to a local minima for a smooth loss function along the lines of, e.g., [33, Theorem 2.1].

**Theorem 6** (AQSGD for smooth nonconvex optimization). *Let $f : \Omega \to \mathbb{R}$ denote a possibly nonconvex and $\beta$-smooth function. Let $\mathbf{w}_0 \in \Omega$ denote an initial point, $\overline{\epsilon_Q}$ and $\overline{N_Q}$ be defined as in Theorem 4, $T \in \mathbb{Z}^{>0}$, and $f^* = \inf_{\mathbf{w} \in \Omega} f(\mathbf{w})$. Suppose that Algorithm 1 is executed for $T$ iterations with a learning rate $\alpha < 2/\beta$ on $M$ processors, each with access to independent stochastic gradients of $f$ with a second-moment bound $B$, such that levels are updated $K$ times where $\boldsymbol{\ell}_k$ with variance bound $\epsilon_{Q,k}$ and code-length bound $N_{Q,k}$ is used for $T_k$ iterations. Then there exists a random stopping time $R \in \{0, \ldots, T\}$ such that AQSGD guarantees*

$$\mathbb{E}[\|\nabla f(\mathbf{w}_R)\|^2] \leq \frac{\beta(f(\mathbf{w}_0) - f^*)}{T} + \frac{2(1 + \overline{\epsilon_Q})B}{M}.$$

*In addition, AQSGD requires at most $\overline{N_Q}$ communication bits per iteration in expectation.*

# I  AQSGD with momentum

The update rule for full-precision unified momentum SGD (UMSGD) is given by [34]

$$\begin{aligned}
\mathbf{y}_{t+1} &= \mathbf{w}_t - \alpha g(\mathbf{w}_t) \\
\mathbf{y}_{t+1}^{\ell} &= \mathbf{w}_t - l\alpha g(\mathbf{w}_t) \\
\mathbf{w}_{t+1} &= \mathbf{y}_{t+1} + \mu\big(\mathbf{y}_{t+1}^{\ell} - \mathbf{y}_t^{\ell}\big),
\end{aligned} \tag{45}$$

where $\mathbf{w}_t$ is the current parameter input and $\mu \in [0, 1)$ is the momentum parameter. Note that the heavy-ball method [35] and Nesterov's accelerated gradient method [36] are the special cases of UMSGD obtained by substituting $l = 0$ and $l = 1$ into Eq. (45), respectively.

The steps for data-parallel version of UMSGD are those in Algorithm 1 by replacing Line 9 with an UMSGD update. We have convergence guarantees for adaptively quantized SGD with momentum (AQSGDM) along the lines of, e.g., [34, Theorem 1]. We first establish the convergence guarantees for convex optimization in the following theorem.

**Theorem 7** (AQSGDM for convex optimization). *Let $f : \mathbb{R}^d \to \mathbb{R}$ denote a convex function with $\|\nabla f(\mathbf{w})\| \leq V$ for all $\mathbf{w}$. Let $\mathbf{w}_0$ denote an initial point, $\mathbf{w}^* = \arg\min f(\mathbf{w})$, $\hat{\mathbf{w}}_T = 1/T \sum_{t=0}^{T} \mathbf{w}_t$, and $\overline{\epsilon_Q}$ and $\overline{N_Q}$ be defined as in Theorem 4.*

*Suppose that AQSGDM is executed for $T$ iterations with a learning rate $\alpha > 0$ on $M$ processors, each with access to independent stochastic gradients of $f$ with a second-moment bound $B$, such that levels are updated $K$ times where $\ell_k$ with variance bound $\epsilon_{Q,k}$ and code-length bound $N_{Q,k}$ is used for $T_k$ iterations. Then AQSGDM satisfies*

$$\mathbb{E}[f(\hat{\mathbf{w}}_T)] - \min_{\mathbf{w} \in \Omega} f(\mathbf{w}) \leq \epsilon_\mu^\ell, \tag{46}$$

*where $\epsilon_\mu^\ell = \mu(f(\mathbf{w}_0) - f(\mathbf{w}^*))/((1-\mu)(T+1)) + (1-\mu)\|\mathbf{w}_0 - \mathbf{w}^*\|^2/(2\alpha(T+1)) + \alpha(1+2l\mu)(V^2 + (1+\overline{\epsilon_Q})B/M)/(2(1-\mu))$.*

*In addition, AQSGD requires at most $\overline{N_Q}$ communication bits per iteration in expectation.*

On nonconvex problems, (weaker) convergence guarantees can be established for AQSGDM. In particular, AQSGDM is guaranteed to converge to a local minima for smooth general loss functions.

**Theorem 8** (AQSGDM for smooth nonconvex optimization). *Let $f : \mathbb{R}^d \to \mathbb{R}$ denote a possibly nonconvex and $\beta$-smooth function with $\|\nabla f(\mathbf{w})\| \leq V$ for all $\mathbf{w}$. Let $\mathbf{w}_0$ denote an initial point, $\mathbf{w}^* = \arg\min f(\mathbf{w})$, and $\overline{\epsilon_Q}$ and $\overline{N_Q}$ be defined as in Theorem 4.*

*Suppose that AQSGDM is executed for $T$ iterations with $\alpha = \min\{(1-\mu)/(2\beta), C/\sqrt{T+1}\}$ for some $C > 0$ on $M$ processors, each with access to independent stochastic gradients of $f$ with a second-moment bound $B$, such that levels are updated $K$ times where $\ell_k$ with variance bound $\epsilon_{Q,k}$ and code-length bound $N_{Q,k}$ is used for $T_k$ iterations. Then AQSGDM satisfies*

$$\min_{t=0,\dots,T} \mathbb{E}[\|\nabla f(\mathbf{w}_t)\|^2] \leq \frac{2(f(\mathbf{w}_0) - f(\mathbf{w}^*))(1-\mu)}{\alpha(T+1)} + \frac{C\tilde{V}}{(1-\mu)^3\sqrt{T+1}},$$

*where*

$$\tilde{V} = \beta\big(\mu^2((1-\mu)l - 1)^2 + (1-\mu)^2\big)(V^2 + (1+\overline{\epsilon_Q})B/M).$$

*In addition, AQSGD requires at most $\overline{N_Q}$ communication bits per iteration in expectation.*

## J    Theoretical guarantees for levels with symmetry

We first obtain variance upper bound for the symmetrical $\boldsymbol{\ell} = [-\ell_{s+1}, \dots, -\ell_1, \ell_1, \dots, \ell_{s+1}]^\top$.

**Theorem 9** (Variance bound). *Let $\mathbf{v} \in \mathbb{R}^d$ and $q \geq 1$. The quantization of $\mathbf{v}$ under $L^q$ normalization satisfies $\mathbb{E}[Q_{\boldsymbol{\ell}}(\mathbf{v})] = \mathbf{v}$. Furthermore, we have*

$$\mathbb{E}[\|Q_{\boldsymbol{\ell}}(\mathbf{v}) - \mathbf{v}\|_2^2] \leq \epsilon_Q \|\mathbf{v}\|_2^2, \tag{47}$$

*where $\epsilon_Q = \ell_1^2 d^{\frac{2}{\min\{q,2\}}} + \frac{(\ell_{j^*+1}/\ell_{j^*} - 1)^2}{4(\ell_{j^*+1}/\ell_{j^*})}$ where $j^* = \arg\max_{1 \leq j \leq s} \ell_{j+1}/\ell_j$.*

*Proof.* Following Proposition 3, the variance is given by

$$\mathbb{E}[\|Q_{\boldsymbol{\ell}}(\mathbf{v}) - \mathbf{v}\|_2^2] = \|\mathbf{v}\|_q^2 \Big( \sum_{r_i \in [0,\ell_1]} (\ell_1^2 - r_i^2) + \sum_{j=1}^s \sum_{r_i \in [\ell_j, \ell_{j+1}]} (r_i - \ell_j)(\ell_{j+1} - r_i) \Big).$$

Note that $\ell_1^2 - r^2 \leq \ell_1^2$ for $r \in [0, \ell_1]$. The rest of the proof follows the proof of Theorem 2. $\qquad\square$

In order to implement an efficient lossless prefix code, we need to know the probabilities associated with our symbols to be coded, *i.e.,* $\{-\ell_{s+1}, \dots, -\ell_1, \ell_1, \dots, \ell_{s+1}\}$. We can obtain those probabilities using the marginal PDF of normalized coordinates:

**Proposition 8.** *Suppose $p(-\theta) = p(\theta)$. The probability of occurrence of $\ell_j$ (weight of symbol $\ell_j$) $\Pr(\ell_j)$ is equal to $\Pr(-\ell_j)$, given by*

$$\Pr(\ell_j) = \int_{\ell_{j-1}}^{\ell_j} \frac{\theta - \ell_{j-1}}{\ell_j - \ell_{j-1}} \, dF(\theta) + \int_{\ell_j}^{\ell_{j+1}} \frac{\ell_{j+1} - \theta}{\ell_{j+1} - \ell_j} \, dF(\theta)$$

*for $j = 2, \ldots, s$. In addition, we have*

$$\Pr(\ell_1) = \Pr(-\ell_1) = \int_{-\ell_1}^{\ell_1} \frac{\theta + \ell_1}{2\ell_1} \, \mathrm{d}F(\theta) + \int_{\ell_1}^{\ell_2} \frac{\ell_2 - \theta}{\ell_2 - \ell_1} \, \mathrm{d}F(\theta),$$

$$\Pr(\ell_{s+1} = 1) = \Pr(-\ell_{s+1}) = \int_{\ell_s}^{1} \frac{\theta - \ell_s}{1 - \ell_s} \, \mathrm{d}F(\theta).$$

In the special case of truncated normal distribution, we have the symbol probabilities in closed-form:

**Corollary 4.** *Suppose normalized coordinates have truncated normal distribution with PDF $p_{\mathcal{T}}$ and CDF $F_{\mathcal{T}}$ defined in Appendix A.2. The probability of occurrence of $\ell_j$ (weight of symbol $\ell_j$) is given by*

$$\Pr(\ell_j) = \Pr(-\ell_j) = \frac{\sigma^2(p_{\mathcal{T}}(\ell_{j-1}) - p_{\mathcal{T}}(\ell_j)) + (\mu - \ell_{j-1})(F_{\mathcal{T}}(\ell_j) - F_{\mathcal{T}}(\ell_{j-1}))}{\ell_j - \ell_{j-1}}$$
$$+ \frac{\sigma^2(p_{\mathcal{T}}(\ell_{j+1}) - p_{\mathcal{T}}(\ell_j)) + (\ell_{j+1} - \mu)(F_{\mathcal{T}}(\ell_{j+1}) - F_{\mathcal{T}}(\ell_j))}{\ell_{j+1} - \ell_j}$$

*for $j = 2, \ldots, s$. In addition, we have*

$$\Pr(\ell_1) = \Pr(-\ell_1) = \frac{\sigma^2(p_{\mathcal{T}}(\ell_2) - p_{\mathcal{T}}(\ell_1)) + (\ell_2 - \mu)(F_{\mathcal{T}}(\ell_2) - F_{\mathcal{T}}(\ell_1))}{\ell_2 - \ell_1}$$
$$+ \frac{(\mu + \ell_1)(F_{\mathcal{T}}(\ell_1) - F_{\mathcal{T}}(-\ell_1))}{2\ell_1},$$

$$\Pr(\ell_{s+1} = 1) = \Pr(-\ell_{s+1}) = \frac{\sigma^2(p_{\mathcal{T}}(\ell_s) - p_{\mathcal{T}}(1)) + (\mu - \ell_s)(F_{\mathcal{T}}(1) - F_{\mathcal{T}}(\ell_s))}{1 - \ell_s}.$$

Finally, we have the following bound on the expected number of communication bits per iteration for quantizing with symmetrical levels.

**Theorem 10** (Code-length bound). *Let $\mathbf{v} \in \mathbb{R}^d$ and $q \geq 1$. The expectation $\mathbb{E}[|\mathrm{ENCODE}(\mathbf{v})|]$ of the number of communication bits needed to transmit $Q_{\boldsymbol{\ell}}(\mathbf{v})$ under $L^q$ normalization is bounded by*

$$\mathbb{E}[|\mathrm{ENCODE}(\mathbf{v})|] \leq b + d(H(L) + 1) \leq b + d(\log_2(2s + 2) + 1), \tag{48}$$

*where $b$ is a constant and $L$ is a random variable with the probability mass function given by Proposition 8.*

## K   Experimental details and additional experiments

In this section, we provide additional experiments for the methods evaluated in Section 5. In addition to baselines discussed in Section 5, we present results for ALQ-N, which minimizes the expected normalized variance using coordinate descent in (3), ALQ with norm adjustments in Section 3.4, ALQ adapted using gradient descent in Section 3.2 (ALQG, ALQG-N), AMQ-N, and AMQ. This section includes full ImageNet runs. Figs. 3a, 3b, 5a and 5b have also been extended to include all variations of the proposed algorithms and baselines.

An implementation challenge is that the value of the statistics, especially the variance, can become very small. This makes PDF and CDF calculations challenging. The challenge is that the value of PDF is very close to zero when it is far from the mean but not exactly zero. In order to overcome this challenge, we use histograms to model the distribution of gradients as a weighted sum of truncated normals. Another problem is the large number of statistics that are calculated. As presented in Section 3, we sample a number of gradients and then normalize the gradients. Then we split the gradients into buckets and calculate average, variance, and norm of each of the buckets. The number of means, variances, and norms can become very large with large networks and small bucket sizes. To reduce computational complexity of the algorithm, we sample uniformly from these values. This number of samples is equal to 20 for small networks such as ResNet-8 and networks trained on CIFAR-10; however, in experiments on ImageNet, we used 350 samples to achieve the desired accuracy.

**Table 3:** Training Hyper-parameters for CIFAR-10 and ImageNet

| Hyperparameter | ResNet-32 on CIFAR-10 | ResNet-110 on CIFAR-10 | ImageNet |
|---|---|---|---|
| Learning Rate | 0.1 | 0.1 | 0.1 |
| LR Decay Schedule | At 45K & 60K | At 45K & 60K | At 300K & 450K |
| Batch Size | 128 | 64 | 64 |
| Momentum | 0.9 | 0.9 | 0.9 |
| Total Iterations | 80K | 80K | 600K |
| Weight Decay | $10^{-4}$ | $10^{-4}$ | $10^{-4}$ |
| Optimizer | SGD | SGD | SGD |

**Table 4:** Validation Accuracy on Full ImageNet Run

| Quantization Method | ResNet-18 on ImageNet |
|---|---|
| Bucket Size | 8192 |
| SGD | $64.67\% \pm 0.13$ |
| SuperSGD | $\mathbf{69.85\% \pm 0.05}$ |
| NUQSGD [21, 22] | $35.43\% \pm 0.28$ |
| QSGDinf [20] | $67.48\% \pm 0.08$ |
| TRN [15] | $63.97\% \pm 0.11$ |
| ALQ | $\mathbf{68.65\% \pm 0.10}$ |
| ALQ-N | $\mathbf{68.50\% \pm 0.10}$ |
| AMQ | $67.76\% \pm 0.09$ |
| AMQ-N | $67.96\% \pm 0.10$ |

One other understudied detail in quantizing is how bucketing is performed. In [20, 21], gradient coordinates in each bucket do not exceed the layer size. It means that the gradient coordinates in a bucket do not contain gradient coordinates from the next layer even if the bucket size is not fully utilized. This leads to creation of under-sized buckets that can be problematic for quantization performance. Different tricks are employed to fix this problem. These tricks include transmitting biases or under-sized buckets in full-precision (not that typically biases are main sources of under-sized buckets). In our implementation, we normalize the buckets network-wise and do not consider the layer size as the bucket size boundary. We only transmit the last bucket in full precision if it is smaller than the specified bucket size.

**Update Schedule.** In the ImageNet and CIFAR-10 runs, adaptive level updates are scheduled at 100 and 2000 iterations only once and every 10K iterations. The reason for this schedule is changes in the gradient statistics over the course of training. As shown in Fig. 1, the average variance changes rapidly during the first iterations and then only changes at every learning rate schedule. In practice, we noticed accuracy degradation especially when the levels are not updated during the initial iterations where the average variance is rapidly changing.

**Convergence of level updating.** Fig. 8 shows the expected normalized variance (the objective in (3)) and expected variance (the objective in (9)) during one step of adapting levels. This figure shows that the objective function in (3) is nonconvex and different initializations lead to sub-optimal solutions. ALQG and ALQG-N refer to variations of ALQ using gradient descent instead of coordinate descent.

**Hyperparameters used for training.** Table 3 shows the hyperparameters used for training CIFAR-10 and ImageNet. These are conventional hyperparameters for training ResNet models. SuperSGD is able to replicate the accuracy reported in [28] showing the correct setting for training.

**Validation accuracy on full ImageNet run** Table 4 shows the validation accuracy of full ImageNet runs on ResNet-18. Total number of iterations required for a full ImageNet run is 600K. This table shows that ALQ and ALQ-N are able to outperform QSGDinf by 1% on ImageNet.

**(a)** Expected Normalized Variance  **(b)** Expected Variance

**Figure 8: Convergence** of different level update methods

**(a)** ResNet-32 on CIFAR-10  **(b)** ResNet-110 on CIFAR-10

**Figure 9: Extended Training loss** on CIFAR-10. All methods use 3 quantization bits. Bucket size for ResNet-110 trained on CIFAR-10 is 16384 and for ResNet-32 is 8192.

Similar performance of normalized and unnormalized variations of AMQ and ALQ methods suggests that for given datasets and deep models, the distribution of normalized gradient coordinates can be represented by either of the forumulations in Section 3. In AMQ-N and ALQ-N, $\mu$ and $\sigma$ values for the truncated normal distribution is equal to the average of $\mu$ and $\sigma$ for individual buckets.

Figs. 10a and 10b show an interesting observation for NUQSGD. Although NUQSGD has worse performance in terms of the training loss and average variance compared to all other approaches, it is able to achieve better validation accuracy. This suggests that NUQSGD is able to generalize better in this specific setting. However, this pattern does not repeat when it comes to the ImageNet dataset.

### K.1  Revised Experiments

Figs. 9 to 12 are extended versions of Figs. 3, 4 and 11 figures in the main body. The difference is that they contain more baselines compared to the figures in the main body. Fig. 9 contains the training loss for the experiments on CIFAR-10. It was not possible to include the same figure for ImageNet, because calculating the full training loss on ImageNet takes a very long time.

The expected variance, training loss, and the validation loss for the results presented in Table 2 are shown in Fig. 13. Although ALQ performs better in expected variance and training loss, it seems to have trouble when it comes to the validation loss for 32-GPUs. We suspect that this is due to the large total batch size used for these experiment that results in overfitting. The batch size for each GPU is 128.

### K.2  Effect of Using Gradient Clipping

TRN [15] introduced the idea of gradient clipping before quantization. Gradient clipping replaces the gradient coordinates that are far from the mean to reduce the gradient variance. The gradient coordinates that are very far from the mean can affect the normalization. In order to tackle this

**(a)** ResNet-32 on CIFAR-10    **(b)** ResNet-110 on CIFAR-10    **(c)** ResNet-18 on ImageNet

**Figure 10: Extended Validation loss** on CIFAR-10 and ImageNet. All methods use 3 quantization bits. Bucket size for ResNet-110 trained on CIFAR-10 is 16384, for ResNet-32 is 8192, and for ResNet-18 on ImageNet is 8192.

**(a)** ResNet-32 on CIFAR-10    **(b)** ResNet-110 on CIFAR-10    **(c)** ResNet-18 on ImageNet

**Figure 11: Extended Variance (no train)** on CIFAR-10 and ImageNet. All methods use 3 quantization bits. Bucket size for ResNet-110 trained on CIFAR-10 is 16384, for ResNet-32 is 8192, and for ResNet-18 on ImageNet is 8192.

problem, they clip the gradients before quantization. The clipping process can be described using Eq. (49):

$$f(g_i) = \begin{cases} g_i & |g_i| \leq c\sigma \\ \text{sign}(g_i).c\sigma & |g_i| > c\sigma \end{cases} \tag{49}$$

The constant $c$ used in TRN equals to 2.5. In order to investigate the effect of gradient clipping in ALQ and AMQ, we train a ResNet-8 on CIFAR-10 dataset for various bucket sizes. Fig. 14 shows the validation accuracy of the baselines and the algorithms we proposed. ALQ and ALQ-N always maintain better or equal accuracy compared to the other quantization schemes. It is also worth noting that the quantization is performed by each layer instead of a performing the quantization across the network without considering the layers.

**Figure 14:** Validation Loss

## K.3   Timing Overhead

In this section, we provide the timing results per step for training ResNet-18 (Table 6) and ResNet-50 (Table 7) on ImageNet with mini-batch size 512. The training setup consists of 4 AWS nodes with one V100 GPU on each. Network bandwidth programmatically constrained to 1GBit/s.

**(a)** ResNet-32 on CIFAR-10     **(b)** ResNet-110 on CIFAR-10     **(c)** ResNet-18 on ImageNet

**Figure 12: Extended Variance** on CIFAR-10 and ImageNet. All methods use 3 quantization bits. Bucket size for ResNet-110 trained on CIFAR-10 is 16384, for ResNet-32 is 8192, and for ResNet-18 on ImageNet is 8192.

**(a)** Variance     **(b)** Training Loss     **(c)** Validation Loss

**Figure 13:** Using 32-GPUs to train ResNet-32 on CIFAR-10.

**Table 5:** Training ResNet50 on ImageNet with min-batch size 512. Time per step for training with 32bits full-precision is 1.2s and with 16 bits full-precision is 0.61s.

| Bits | Bucket size | Time per step (s) | Ratio to FP32 | Ratio to FP16 |
|------|-------------|-------------------|---------------|---------------|
| 2 | 64 | 0.41 | 0.34 | 0.67 |
| 2 | 256 | 0.39 | 0.33 | 0.64 |
| 2 | 1024 | 0.38 | 0.32 | 0.62 |
| 2 | 8192 | 0.38 | 0.32 | 0.62 |
| 2 | 16384 | 0.38 | 0.32 | 0.62 |
| 3 | 64 | 0.4 | 0.33 | 0.66 |
| 3 | 256 | 0.4 | 0.33 | 0.66 |
| 3 | 1024 | 0.4 | 0.33 | 0.66 |
| 3 | 8192 | 0.4 | 0.33 | 0.66 |
| 3 | 16384 | 0.38 | 0.32 | 0.62 |
| 4 | 64 | 0.42 | 0.35 | 0.69 |
| 4 | 256 | 0.41 | 0.34 | 0.67 |
| 4 | 1024 | 0.4 | 0.33 | 0.66 |
| 4 | 8192 | 0.4 | 0.33 | 0.66 |
| 4 | 16384 | 0.4 | 0.33 | 0.66 |
| 5 | 64 | 0.43 | 0.36 | 0.70 |
| 5 | 256 | 0.42 | 0.35 | 0.69 |
| 5 | 1024 | 0.41 | 0.34 | 0.67 |
| 5 | 8192 | 0.4 | 0.33 | 0.66 |
| 5 | 16384 | 0.4 | 0.33 | 0.66 |
| 6 | 64 | 0.42 | 0.35 | 0.69 |
| 6 | 256 | 0.41 | 0.34 | 0.67 |
| 6 | 1024 | 0.41 | 0.34 | 0.67 |
| 6 | 8192 | 0.41 | 0.34 | 0.67 |
| 6 | 16384 | 0.41 | 0.34 | 0.67 |
| 7 | 64 | 0.45 | 0.38 | 0.74 |
| 7 | 256 | 0.43 | 0.36 | 0.70 |
| 7 | 1024 | 0.42 | 0.35 | 0.69 |
| 7 | 8192 | 0.42 | 0.35 | 0.69 |
| 7 | 16384 | 0.43 | 0.36 | 0.70 |
| 8 | 64 | 0.45 | 0.38 | 0.74 |
| 8 | 256 | 0.44 | 0.37 | 0.72 |
| 8 | 1024 | 0.43 | 0.36 | 0.70 |
| 8 | 8192 | 0.43 | 0.36 | 0.70 |
| 8 | 16384 | 0.43 | 0.36 | 0.70 |

**Table 6:** Training ResNet18 on ImageNet with min-batch size 512. Time per step for training with 32bits full-precision is 0.57s and with 16 bits full-precision is 0.28s.

| Bits | Bucket size | Time per step | Ratio to FP32 | Ratio to FP16 |
|------|-------------|---------------|---------------|---------------|
| 2 | 64 | 0.13 | 0.23 | 0.46 |
| 2 | 256 | 0.12 | 0.21 | 0.43 |
| 2 | 1024 | 0.11 | 0.19 | 0.39 |
| 2 | 8192 | 0.11 | 0.19 | 0.39 |
| 2 | 16384 | 0.11 | 0.19 | 0.39 |
| 3 | 64 | 0.13 | 0.23 | 0.46 |
| 3 | 256 | 0.12 | 0.21 | 0.43 |
| 3 | 1024 | 0.12 | 0.21 | 0.43 |
| 3 | 8192 | 0.12 | 0.21 | 0.43 |
| 3 | 16384 | 0.12 | 0.21 | 0.43 |
| 4 | 64 | 0.13 | 0.23 | 0.46 |
| 4 | 256 | 0.13 | 0.23 | 0.46 |
| 4 | 1024 | 0.12 | 0.21 | 0.43 |
| 4 | 8192 | 0.12 | 0.21 | 0.43 |
| 4 | 16384 | 0.12 | 0.21 | 0.43 |
| 5 | 64 | 0.13 | 0.23 | 0.46 |
| 5 | 256 | 0.13 | 0.23 | 0.46 |
| 5 | 1024 | 0.13 | 0.23 | 0.46 |
| 5 | 8192 | 0.13 | 0.23 | 0.46 |
| 5 | 16384 | 0.13 | 0.23 | 0.46 |
| 6 | 64 | 0.14 | 0.25 | 0.50 |
| 6 | 256 | 0.13 | 0.23 | 0.46 |
| 6 | 1024 | 0.13 | 0.23 | 0.46 |
| 6 | 8192 | 0.13 | 0.23 | 0.46 |
| 6 | 16384 | 0.13 | 0.23 | 0.46 |
| 7 | 64 | 0.14 | 0.25 | 0.50 |
| 7 | 256 | 0.13 | 0.23 | 0.46 |
| 7 | 1024 | 0.14 | 0.25 | 0.50 |
| 7 | 8192 | 0.13 | 0.23 | 0.46 |
| 7 | 16384 | 0.13 | 0.23 | 0.46 |
| 8 | 64 | 0.15 | 0.26 | 0.54 |
| 8 | 256 | 0.14 | 0.25 | 0.50 |
| 8 | 1024 | 0.14 | 0.25 | 0.50 |
| 8 | 8192 | 0.14 | 0.25 | 0.50 |
| 8 | 16384 | 0.14 | 0.25 | 0.50 |

**Table 7:** Additional overhead of proposed methods for training ResNet18 on ImageNet (Table 6). We also show the cost of performing 3 updates relative to the total cost of training for 60 epochs. Full-precision training for 60 epochs with 32 bits takes 95 hours while with 16 bits takes 46 hours.

| Bits | Bucket size | Quantization Method | Time per update | Ratio to FP32 | Ratio to FP16 |
|------|-------------|---------------------|-----------------|---------------|---------------|
| 3 | 64 | ALQ-N | 1012 | 0.89 | 1.81 |
| 3 | 256 | ALQ-N | 630 | 0.55 | 1.13 |
| 3 | 1024 | ALQ-N | 533 | 0.47 | 0.95 |
| 3 | 8192 | ALQ-N | 559 | 0.49 | 1.00 |
| 3 | 16384 | ALQ-N | 591 | 0.52 | 1.06 |
| 4 | 64 | ALQ-N | 1170 | 1.03 | 2.09 |
| 4 | 256 | ALQ-N | 822 | 0.72 | 1.47 |
| 4 | 1024 | ALQ-N | 733 | 0.64 | 1.31 |
| 4 | 8192 | ALQ-N | 681 | 0.60 | 1.22 |
| 4 | 16384 | ALQ-N | 684 | 0.60 | 1.22 |
| 6 | 64 | ALQ-N | 2036 | 1.79 | 3.64 |
| 6 | 256 | ALQ-N | 1710 | 1.50 | 3.05 |
| 6 | 1024 | ALQ-N | 1556 | 1.36 | 2.78 |
| 6 | 8192 | ALQ-N | 1574 | 1.38 | 2.81 |
| 6 | 16384 | ALQ-N | 1671 | 1.47 | 2.98 |
| 8 | 64 | ALQ-N | 5604 | 4.92 | 10.01 |
| 8 | 256 | ALQ-N | 5253 | 4.61 | 9.38 |
| 8 | 1024 | ALQ-N | 5478 | 4.81 | 9.78 |
| 8 | 8192 | ALQ-N | 5180 | 4.54 | 9.25 |
| 8 | 16384 | ALQ-N | 5576 | 4.89 | 9.96 |
| 3 | 64 | ALQ | 1032 | 0.91 | 1.84 |
| 3 | 256 | ALQ | 585 | 0.51 | 1.04 |
| 3 | 1024 | ALQ | 444 | 0.39 | 0.79 |
| 3 | 8192 | ALQ | 474 | 0.42 | 0.85 |
| 3 | 16384 | ALQ | 477 | 0.42 | 0.85 |
| 4 | 64 | ALQ | 930 | 0.82 | 1.66 |
| 4 | 256 | ALQ | 529 | 0.46 | 0.94 |
| 4 | 1024 | ALQ | 450 | 0.39 | 0.80 |
| 4 | 8192 | ALQ | 431 | 0.38 | 0.77 |
| 4 | 16384 | ALQ | 486 | 0.43 | 0.87 |
| 6 | 64 | ALQ | 974 | 0.85 | 1.74 |
| 6 | 256 | ALQ | 573 | 0.50 | 1.02 |
| 6 | 1024 | ALQ | 489 | 0.43 | 0.87 |
| 6 | 8192 | ALQ | 428 | 0.38 | 0.76 |
| 6 | 16384 | ALQ | 438 | 0.38 | 0.78 |
| 8 | 64 | ALQ | 1051 | 0.92 | 1.88 |
| 8 | 256 | ALQ | 637 | 0.56 | 1.14 |
| 8 | 1024 | ALQ | 516 | 0.45 | 0.92 |
| 8 | 8192 | ALQ | 508 | 0.45 | 0.91 |
| 8 | 16384 | ALQ | 516 | 0.45 | 0.92 |