[Reviews · NeurIPS 2020]

Review 1

Summary and Contributions: The paper proposes an adaptive quantization of gradients on the fly to reduce communication in distributed neural network training. Unlike quantization under fixed discrete values in previous work, the paper optimizes those discrete values on the fly based on the dynamic statistics of the gradient distribution. The problem formulation is sound and the method is well theoretically studied. However, the experiments and comparison should be improved to solidify this paper.

Strengths: 1. the method is well motivated (optimized quantization values and dynamic update of those values when the gradient distribution shifts); 2. theoretical study is sound.

Weaknesses: The most significant limitation of this work is its experiment part. The following aspects should be fixed to support the claim: 1. why there is a large gap between SGD and SuperSGD? Generally, the accuracy should be close whenever the model is trained in a single GPU or multiple GPUs. Was it because different total mini-batch sizes were used? 2. clarify if/how weights are communicated through the network, and how gradients are exchanged. 3. In Figure 7(a), the validation accuracy drops when bucket size is very small. This is controversial. When bucket size is smaller, the variance is smaller, and thus the accuracy should be closer to full-precision SGD. In the extreme case when bucket size is 1, quantized SGD is floating-point SGD. In Figure 7(b), why the TRN line is a flat when increasing the bit-width. 4. putting TRN in the content of "3 quantization bits" is misleading. The TRN only used "ternary levels" (three levels) as stated in Line 209. Please clarify this in Table 1, Figure 3, Figure 4 and Figure 7(b). Such as "All methods use 3 quantization bits". Also please clarify if the "gradient clipping" in TRN was used or not in the implementation. "gradient clipping" is also an adaptive processing to reduce the variance of gradients by limiting the maximum norm.

Correctness: See Weaknesses.

Clarity: 1. explain P_L in equation (6) 2. explain x- and y-axis in Figure 6. 3. some markers are missing in Figure 7(b), such as, markers at the 3 bits.

Relation to Prior Work: It's generally clearly discussed. "To the best of our knowledge, matching the validation loss of SuperSGD has not been achieved in any previous work using only 3 bits": please clarify the condition. At least, TRN "using only 3-level gradients achieves 0.92% top-1 accuracy improvement for AlexNet" [15].

Reproducibility: Yes

Additional Feedback: 1. Evaluating the method in a larger scale distributed systems with much more nodes (>>4 nodes) may boost the gain of this method, as the gradient variance is large at each node when the batch size per node is small. 2. consider to remove markers in Figure 7 for a better view. ---------------------------- The rebuttal resolved part of my concerns regarding their experiments, but the SGD vs SuperSGD issue. Each baseline should have been done with well-tuned hyper-parameters for comparison. This issue plus the reported bug in the code rise/reassure my concerns on the experiment part. Considering the sound theoretical part, my final score will be 6.5 if I can, given all details in the rebuttal are correct and will be incorporated into the final version.


Review 2

Summary and Contributions: The paper proposed a new adaptive quantization scheme to reduce the communication cost during parallel training.

Strengths: The paper is well written and also provides step-by-step math computations.

Weaknesses: -- Why is the performance of SGD/SuperSGD so different? -- Figure 7 uses ResNet-8, which is weird to me since it is never used in pervious examples in the paper. -- Line 243 -- 250, the computation overhead is based on bucketsize=64, however all main results are based on bucketsize=8k/16k -- Though the method is proposed to speed up parallel training, the number nodes (or simulated nodes) is only 4. -- For AMQ, what p are you using? Except p=1/2, other should be hard to implement efficiently in practice. -- The theoretical results are useful but trivial to get. The authors should consider shorten that part. -- Could you compare with your results with other quantization methods, which aim to use better quantization scheme, like LQNet?

Correctness: Yes.

Clarity: Yes.

Relation to Prior Work: Yes.

Reproducibility: Yes

Additional Feedback:


Review 3

Summary and Contributions: The paper studies quantizing gradients in a distributed learning scenario in order to reduce communication burden. Specifically, they learn the quantization levels for both cases of uniform and exponential quantization by minimizing the variance between the quantized and the original values in expectation. They further prove the theoretical bounds for this variance. The authors further experimentally show they can quantize the gradients down to 3 bits without loss of accuracy.

Strengths: The proposed method is established using theoretical guarantees of convergence and derives the necessary communication bounds which can be used for the purpose of comparison with other gradient compression methods.

Weaknesses: Some aspects of the evaluation methodology have not been fully explained. It might be better to provide overall communication load requirements instead of quantization bitwidth for comparison with previous works as they may not compress all gradient values with a uniform bit-width (e.g. pruning).

Correctness: To the best of my understanding the proposed bounds are correct. More details for evaluation of the experimental results would help judge their correctness. It is mentioned that evaluation is done by simulating running on 4 GPUs using a single GPU. Can the authors provide more details about how this simulation was carried out?

Clarity: Yes. The paper is clear and easy to follow.

Relation to Prior Work: The authors have performed comparisons with several previous works using different DNN models and image classification datasets.

Reproducibility: Yes

Additional Feedback:


Review 4

Summary and Contributions: The paper proposes two adaptive quantization techniques for efficient communication in distributed SGD setups. Quantization levels are chosen to minimize the expected (normalized) variance of the gradient and can either be: 1. Exponentially spaced levels controlled by a single multiplier 2. Adaptable individual levels chosen to minimize the variance. The paper provides theoretical worst-case guarantees for the variance and code-length.

Strengths: - Establishes an upper bound on variance and code length. - Empirical results support the claims made in the paper

Weaknesses: - The claim "robust to all values of hyperparameters" looks like it is hard to defend since the experiments are performed only on varying choices of bucket size and number of bits. What about momentum, batch size etc.

Correctness: I did not read the proofs for the theorems so cannot comment on these. However, the empirical methodology supports the main claims.

Clarity: Yes. Grammar: performs nearly as good as adaptive methods -> performs nearly as well as adaptive methods

Relation to Prior Work: There is discussion of prior work, but discussion of how this papers differs from previous work has to be gleaned.

Reproducibility: Yes

Additional Feedback:

[Author Response · NeurIPS 2020]

We thank the reviewers for their thoughtful comments. Their comments have helped us improve the experiments and
clarity of the paper significantly as we discuss below.

[R1,R3] There is a large gap between SGD and SuperSGD. To clarify, we run both SGD and SuperSGD for the same
number of steps and similar learning rate schedule but different number of GPUs with the same mini-batch size on each
GPU. On ImageNet, SGD has lower accuracy than SuperSGD because the number of steps taken is not enough for
convergence with mini-batch $64$. On CIFAR-10, SGD has $1.5\%$ higher accuracy that as noted in the paper is a known
gap that can be reduced by extensive hyperparameter tuning [29]. We will clarify this in the paper.

[R1] Clarify that TRN uses three levels and it has competitive performance on AlexNet. Thank you for noting gradient
clipping in TRN. In particular, Fig. 1 shows the improvement all methods gain from gradient clipping. We will
also clarify in the revision that TRN uses three levels and it achieves 0.92% accuracy improvement for AlexNet with
mini-batch size of $1024$. We emphasize that our methods match the performance of SuperSGD in more training settings.

[R1] In Fig. 7(a), the validation accuracy should improve when the bucket size is very small. Thank you for noting this
issue. We fixed a bug that only affected the runs with very small bucket sizes. As expected, all validation accuracies
improve uniformly by reducing the bucket size. We will replace the figure in the revision.

[R1,R3] Provide experiments with more number of GPUs. We present new results in Table 1 similar to Table 1 in the
paper but with 16 and 32 GPUs. Proposed methods ALQ and ALQ-N perform significantly better than prior works and
reach the performance of SGD that has converged. As noted above, the SuperSGD is slightly worse than SGD.

[R3] Computation overhead is based on bucketsize=64, however all main results are based on bucketsize=8k/16k. On
ImageNet, we save at least 60 hours from 95 hours of training and add only an additional cost of at most 10 minutes in
total to adapt quantization. For bucket sizes 8192 and 16384 used in the paper and 3-8 bits, the per step cost relative to
SuperSGD (32-bits) is between 21% to 25% for ResNet-18 on ImageNet and 32% to 36% for ResNet-50. That is the
same as the cost of NUQSGD and QSGDinf without additional coding or pruning with the same number of bits and
bucket sizes. The cost of the additional update specific to ALQ is between 0.4% and 0.5% of the total training time. We
will provide a full table of timing results in the revision with varying bucket sizes and bits.

[R4] How about comparing different methods in terms of overall communication load? For fair comparison, none of the
methods use an additional coding scheme or pruning to further reduce communication costs.

[R5] Do authors claim that ALQ/AMQ are robust to all values of momentum, batch size, etc? No. Our robustness
claim refers only to the bucket size and number of bits. We have justified this claim by comparing the performance of
methods across a wide range of bucket sizes and number of bits. We will clarify this in the revision.

[R3] How about shortening theoretical results? In the revision, we move the details of Theorem 4 to the appendix.

[R1] How are gradients/weights communicated through the network? We consider a synchronous setting for sharing
gradients similar to [20] (we do not communicate weights).

[R5,R3] Elaborate on Related Work. How about LQ-Net? We will discuss expand related works in revision. The goal
in LQ-Net is to quantize weights and activations such that the inner products of them can be computed efficiently using
bitwise operations (single-processor setting). Compared to LQ-Net, our schemes are more efficient and do not need
additional memory for encoding vectors. In training, LQ-Net updates levels for each vector to be quantized.

[R4]: Details of simulation setting. Appendix K describes the setting which we will expand on. We encourage reviewers
to see the code.

[R3]: Can AMQ with $p$ beyond $\frac{1}{2}$ be implemented efficiently? Selecting $p = \frac{1}{2}$ might simplify bitwise operations.
However, as our experiments show, $p = \frac{1}{2}$ is far from optimal and the computational overhead of AMQ is negligible.

| Method | 16 GPUs | 32 GPUs |
|---|---|---|
| SGD | **92.40% $\pm$ 0.06** | **92.47% $\pm$ 0.09** |
| SuperSGD | **92.17% $\pm$ 0.08** | 92.19% $\pm$ 0.04 |
| NUQSGD | 85.82% $\pm$ 0.03 | 86.36% $\pm$ 0.01 |
| QSGDinf | 89.61% $\pm$ 0.03 | 89.81% $\pm$ 0.05 |
| TRN | 88.68% $\pm$ 0.10 | 90.22% $\pm$ 0.05 |
| ALQ | **91.91% $\pm$ 0.06** | **91.89% $\pm$ 0.07** |
| ALQ-N | **92.07% $\pm$ 0.04** | **91.83% $\pm$ 0.03** |
| AMQ | 91.58% $\pm$ 0.05 | 91.38% $\pm$ 0.06 |
| AMQ-N | 91.41% $\pm$ 0.08 | 91.40% $\pm$ 0.02 |

**Table 1:** Validation accuracy of ResNet32 on CIFAR-10 using 3 quantization bits (except for SGD, SuperSGD, and TRN) and bucket size 16384.

**Figure 1:** Effect of bucket size with gradient clipping from TRN on training ResNet8 on CIFAR-10.

[Meta-Review · NeurIPS 2020]

All of the reviews are relatively short, and largely focus on questions about the experiment results, for which the author rebuttal give reasonable explanations. I think the main strength of the paper is its analytical approach for adaptive quantization based on changing statistics of the stochastic gradients, which some reviewers pointed out but didn't give substantive comments or discussion. These is a large concurrent literature on using gradient quantization in distributed training of large machine learning models. The adaptive quantization methods presented in this paper is well motivated, the technical treatment is novel and has sound theoretical analysis on the variance bound and code length (Theorems 1-3). The consequences on the optimization algorithm, namely AQSGD, are direct consequences of the variance bounds and the convergence results in Theorem 4 looks standard. The empirical study is well designed, although there are concerns on particular revelations and the way they are conducted (simulated using a single GPU instead of true distributed computation). Overall I believe this work has sufficient novelty, with a fresh theoretical contribution to quantization-based distributed optimization, and may have practical impact as well. Therefore I recommend acceptance. The writing of the paper is clear in general. However, there are a few places where the notations are not well introduces and reference of critical formulas in the supplementary materials. Here are a few examples: * Theorem 2 and Theorem 3 mention "under L^q normalization" which is not found anywhere in the paper, and even vague in the appendix where it is proved. * Theorem 3 refer to "L is a random variable with probability mass function given by Proposition 6," but that proposition is in the appendix. I understand that there are space limitations, but such omissions leave the statement meaningless without referring to the appendix. The authors are expected to make careful revisions to address these issues and the questions raised by the reviewers on the empirical results.